# AgentNet: Decentralized Evolutionary Coordination for LLM-based Multi-Agent Systems

**Yingxuan Yang**[1]*, **Huacan Chai**[1]*, **Shuai Shao**[1],
**Yuanyi Song**[1], **Siyuan Qi**[1], **Renting Rui**[1], **Weinan Zhang**[1,2]†
[1]Shanghai Jiao Tong University, [2]Shanghai Innovation Institute
{zoeyyx, fatcat, wnzhang}@sjtu.edu.cn

## Abstract

The rapid advancement of large language models (LLMs) has enabled the development of multi-agent systems where multiple LLM-based agents collaborate on complex tasks. However, existing systems often rely on centralized coordination, leading to scalability bottlenecks, reduced adaptability, and single points of failure. Privacy and proprietary knowledge concerns further hinder cross-organizational collaboration, resulting in siloed expertise. We propose **AgentNet**, a decentralized, Retrieval-Augmented Generation (RAG)-based framework that enables LLM-based agents to specialize, evolve, and collaborate autonomously in a dynamically structured Directed Acyclic Graph (DAG). Unlike prior approaches with static roles or centralized control, AgentNet allows agents to adjust connectivity and route tasks based on local expertise and context. AgentNet introduces three key innovations: (1) a fully decentralized coordination mechanism that eliminates the need for a central orchestrator, enhancing robustness and emergent intelligence; (2) dynamic agent graph topology that adapts in real time to task demands, ensuring scalability and resilience; and (3) a retrieval-based memory system for agents that supports continual skill refinement and specialization. By minimizing centralized control and data exchange, AgentNet enables fault-tolerant, privacy-preserving collaboration across organizations. Experiments show that AgentNet achieves higher task accuracy than both single-agent and centralized multi-agent baselines.

## 1 Introduction

By leveraging collective intelligence through parallel decision-making or workflow collaboration, LLM-based Multi-Agent Systems (MAS) have emerged as a promising framework for tackling complex real-world problems [7, 20, 26, 27]. However, most MAS following the workflow collaboration paradigm rely heavily on a centralized controller or a static, predefined workflow to allocate tasks among agents with fixed roles [2, 9, 22, 25, 29]. While such designs simplify orchestration, they also introduce inherent constraints—including limited scalability, a single point of failure, and challenges to cross-organizational collaboration due to privacy and proprietary knowledge concerns.

A more critical drawback arises from the inability of these systems to adapt to real-time fluctuations in agent performance or rapidly changing task requirements. Relying on a central controller inflates deployment complexity and restricts dynamic role reassignment, rendering the system vulnerable when the controller fails or becomes overloaded. Furthermore, rigid role definitions prevent agents from flexibly leveraging their full expertise in dynamic environments, ultimately undermining both efficiency and scalability. Taken together, these limitations highlight the need for more decentralized,

---

*These authors contributed equally to this work.
†Weinan Zhang is the corresponding author.

39th Conference on Neural Information Processing Systems (NeurIPS 2025).

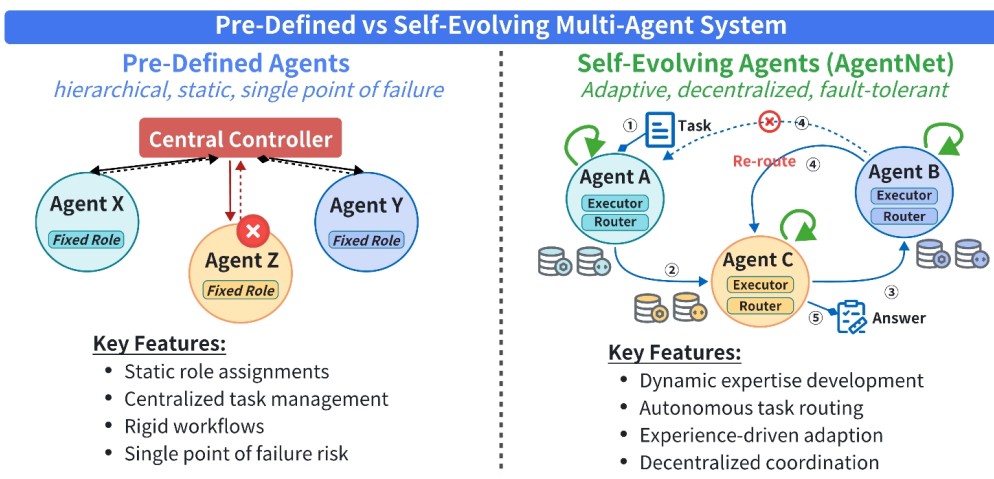

Figure 1: Conceptual comparison between traditional Pre-Defined Multi-Agent Systems and the proposed Self-Evolving AgentNet. While conventional systems rely on hierarchical and static structures with centralized control, AgentNet features adaptive, decentralized coordination and dynamic expertise evolution, enabling robust and scalable performance.

fault-tolerant approaches that support dynamic task allocation, enhance adaptability, and safeguard privacy across organizational boundaries.

Beyond the scalability and failure-tolerance issues previously discussed, centralized architectures become even more problematic when organizations attempt to collaborate at scale [27, 19]. Each institution—be it an enterprise, research lab, or government agency—typically holds proprietary expertise, sensitive data, or both. In a centralized setup, concerns over data ownership, privacy regulations, and inconsistent governance often create barriers that prevent free exchange of knowledge. As a result, LLM-based agents contributed by multiple organizations remain siloed, unable to fully capitalize on each other's specialized capabilities or datasets. This fragmentation not only hampers collective intelligence but also highlights the urgency of developing secure, decentralized collaboration mechanisms. By enabling each participant to maintain and share only the minimal necessary information, these mechanisms address data confidentiality requirements while still allowing for a richer, more collaborative multi-agent ecosystem.

To address these challenges in multi-agent systems, we propose AgentNet, a novel framework designed to foster adaptive agent evolution, optimize task coordination, and preserve privacy. In real-world deployments, each agent in a multi-agent system may have its own role, along with private databases, tools, and other unique resources. The quality and domain of these databases directly influence the agent's capability through training parameters or demonstrations. In AgentNet, we model this reality using a RAG-based mechanism, enabling each agent to maintain distinct knowledge, perform unique updates, and retrieve information effectively. By eliminating the reliance on a central orchestrator, AgentNet enables agents to dynamically reconfigure their connections and redistribute tasks, forming a self-organizing, fault-tolerant architecture. Within this architecture, tasks are efficiently routed via a Directed Acyclic Graph (DAG) [11, 1], which supports flexible collaboration and prevents cyclic dependencies.

Unlike traditional MAS frameworks that fix each agent's role (as shown in Figure 1), **AgentNet** incorporates a retrieval-based RAG [13, 6, 34] memory mechanism to refine agent expertise over time. In real-world deployments, each agent in a multi-agent system may have its own role, along with private databases, tools, and other unique resources. The quality and domain of these private data sources directly influence the agent's capability through training parameters or demonstrations. To model this reality, each agent in AgentNet maintains a limited-capacity pool of successful task trajectories and domain-specific knowledge. When a new task arises, the agent retrieves the most relevant trajectories through few-shot learning to improve its decision-making and adapt its expertise. To prevent memory overflow, agents autonomously prune less pertinent trajectories, ensuring the retention of valuable knowledge. This dynamic specialization strategy not only streamlines task allocation and agent adaptation but also supports a highly scalable and privacy-preserving environment for multi-agent collaboration.

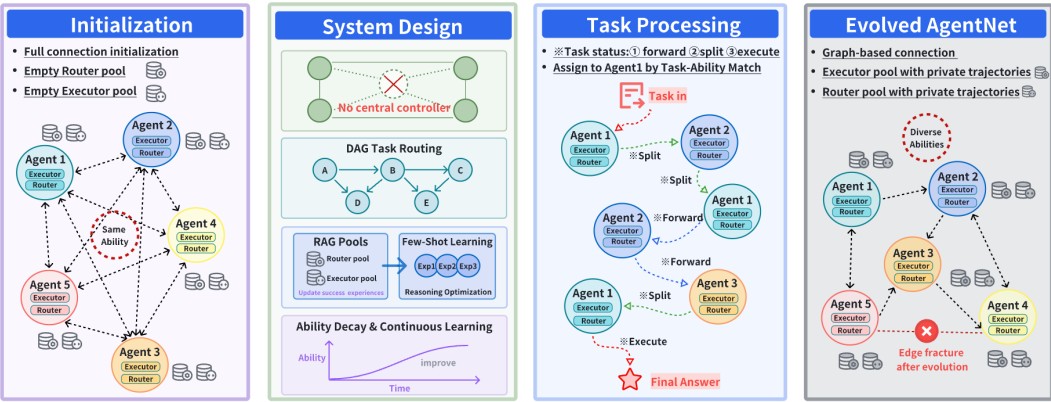

Figure 2: Illutration of AgentNet. Initially, agents are fully connected and equipped with executors and routers. The system eliminates the need for a central controller, using a DAG for dynamic task routing and agents leveraging RAG pools and few-shot learning. In the evolved phase, the network adapts with agents developing private trajectories and diversified abilities.

AgentNet's core design is built upon several key innovations:

- **Fully Decentralized Paradigm**: By removing the need for a central orchestrator, AgentNet fosters emergent collective intelligence. Decision-making authority is distributed across all agents, thereby eliminating single points of failure and allowing each agent to coordinate, delegate, and specialize as conditions evolve. This approach leads to a self-organizing and fault-tolerant architecture that can rapidly respond to new tasks and unforeseen challenges. This decentralized setup also encourages emergent collective intelligence—in other words, agents can collectively discover and refine optimal strategies rather than waiting for instructions from a central controller.

- **Dynamically Evolving Graph Topology**: AgentNet employs a network structure in which both nodes (agents) and edges (agent-to-agent connections) adapt in real time based on task demands and agent performance. Rather than relying on fixed workflows, the system continuously reconfigures its topology to optimize information flow and task distribution, ensuring scalability and resilience in complex, changing environments.

- **Adaptive Learning Mechanism for Expertise Refinement**: AgentNet's third innovation is its retrieval-based memory system, enabling agents to capture and update knowledge from successful task trajectories. This mechanism continuously refines each agent's specialized skills without altering the network's topology, allowing agents to avoid over-reliance on outdated information and sustain high performance in dynamic scenarios.

Moreover, each of these three innovations inherently enhances data privacy. By eliminating a central orchestrator, every agent stores and processes knowledge locally, sharing only minimal task-relevant metadata. The dynamic graph topology further confines data flow to necessary agent-to-agent interactions, reducing the exposure of sensitive information. Meanwhile, the retrieval-based memory mechanism restricts how much and how long data is retained, pruning outdated trajectories so that only high-value knowledge persists. Together, these design choices safeguard privacy and intellectual property, particularly crucial for cross-organizational collaborations.

Our experimental evaluation shows that AgentNet significantly outperforms traditional LLM-based multi-agent frameworks in dynamic environments, demonstrating improved task efficiency, specialization stability, and adaptive learning speed. These results highlight the effectiveness of decentralized evolutionary coordination in large-scale AI ecosystems.

## 2 Related Work

### 2.1 LLM-based Multi-Agent Systems

LLM-based multi-agent systems (LaMAS) have rapidly evolved, with early frameworks like AutoGen [25] and MetaGPT [9] establishing structured, centralized workflows. These systems enabled effective coordination but suffered from scalability issues, single points of failure, and limited adaptability. Subsequent work, such as AgentScope [5] and MegaAgent [22], introduced modular and hierarchical

designs to improve robustness. However, they remain centrally orchestrated and typically rely on single LLMs, with static task workflows that hinder dynamic adaptation. AgentNet departs from this centralized paradigm by introducing a fully decentralized architecture. Agents specialize dynamically, collaborate via a DAG-structured network, and evolve their expertise through retrieval-based memory, enabling scalable, fault-tolerant coordination.

## 2.2 Evolutionary and Adaptive Agent Systems

Inspired by biological evolution, several frameworks optimize agent behaviors through prompt evolution [4, 12], topology adaptation [35, 15], and role specialization [3, 16]. While promising, most operate under centralized control and focus on individual agents rather than system-level decentralization. Recent advances such as AgentSquare [18] and EvoMAC [10] explore automated workflow design and self-adaptive strategies, yet often lack mechanisms for scalable, decentralized coordination. AgentNet addresses this gap by combining evolutionary learning with decentralized control. It enables heterogeneous agents to adapt roles and strategies in real time, supporting scalable and dynamic collaboration across large agent networks.

# 3 Methodology

## 3.1 Preliminary of AgentNet

Unlike traditional MAS frameworks with fixed agent roles and rigid workflows using central co-ordinators, AgentNet creates a privacy-preserving, collective intelligence multi-agent system with high scalability and failure-tolerance by leveraging an innovative framework, consisting of a fully decentralized network architecture, a dynamic task allocation mechanism, and an adaptive agent learning method, as illustrated in Figure 2.

We begin with a brief introduction of AgentNet, including notation and basic architectures of agents employed. Formally, we define AgentNet as a tuple $\mathcal{G} = (\mathcal{A}, \mathcal{E})$, where $\mathcal{A} = \{a_1, a_2, ..., a_n\}$ represents the set of autonomous agents, $\mathcal{C} = \{c_1, c_2, ..., c_n\}$ represents each agent's ability, and $\mathcal{E} \subseteq \mathcal{A} \times \mathcal{A}$ represents the communication connections between agents, specifically $e_{i,j} \in \mathcal{E}$ referring to a unidirectional connection from Agent $a_i$ to Agent $a_j$. For each agent $a_i \in \mathcal{A}$ contains two key components. $rou_i$ is an agent router, responsible for analyzing received routing queries and making routing decisions. $exe_i$ is an agent executor, responsible for responding to executing queries through operations and tools. The two components mentioned above are underpinned by a substantial LLM that leverages its extensive knowledge and understanding to solve specific problems. Furthermore, both $rou_i$ and $exe_i$ in $a_i$ maintain fixed-size memory modules $\mathbb{M}_i^{rou}$ and $\mathbb{M}_i^{exe}$, respectively, providing $a_i$ with powerful adaptive evolutionary capabilities by storing and utilizing the agent's experiences through the RAG mechanism.

For optimization, AgentNet will be given a series of tasks denoted as $\boldsymbol{T} = \{t_1, t_2, ..., t_M\}$ to resolve, along with an evaluation function $Eval(\cdot)$. The optimization goal of AgentNet is to maximize the evaluated score by $Eval(\cdot)$ for the solution output by AgentNet, specifically optimizing $\mathcal{A}$ and $\mathcal{E}$, as the following formula:

$$\mathcal{G}^* = (\mathcal{A}^*, \mathcal{E}^*) = \arg\max_{\mathcal{A}, \mathcal{E}} \; Eval(\mathcal{G}, \boldsymbol{T}). \tag{1}$$

## 3.2 Decentralized Network Topology

Mathematically, we represent the architecture of AgentNet as $\mathcal{G}_m = (\mathcal{A}_m, \mathcal{E}_m)$ when given the $m + 1$-th task $t_{m+1}$ after completing task $t_m$, where $\mathcal{A}_m = \{a_1^m, a_2^m, ..., a_n^m\}$ represents the states of agents after task $t_m$ and $\mathcal{E}_m \subseteq \mathcal{A}_m \times \mathcal{A}_m$ represents the set of directed edges between agents and each edge $e_{i,j}^m$ means a directed edge from $a_i^m$ to $a_j^m$. A weight matrix $w_m$ will be maintained throughout all the tasks before $t_{m+1}$ to weight the connection between agents, namely $w_m(i, j)$. After completing $t_{m+1}$, $w_{m+1}$ can be updated using the following formula from $w_m$:

$$w_{m+1}(i, j) = \alpha \cdot w_m(i, j) + (1 - \alpha) \cdot S(a_i^{m+1}, a_j^{m+1}, t_{m+1}), \tag{2}$$

where $\alpha \in [0, 1]$ is a decay factor that balances historical performance with recent interactions, and $S(a_i^{m+1}, a_j^{m+1}, t_{m+1})$ is a success metric for task $t_{m+1}$ routed from agent $a_i^{m+1}$ to $a_j^{m+1}$. This adaptive weighting mechanism ensures that the network continuously refines its structure based on operational experience.

As shown in Figure 3, AgentNet employs a dual-role architecture where each agent $a_i$ consists of a router $rou_i$ for task distribution and an executor $exe_i$ for task execution. Details about these modules will be discussed in the following sections. Crucially, the router enables AgentNet's decentralized structure, as each agent independently makes routing decisions without relying on any centralized coordinator. This decentralized approach contrasts traditional LLM-based multi-agent systems, which typically rely on a central controller for task allocation. In AgentNet, agents autonomously decide task routing based on local information and specific task requirements. This ensures distributed decision-making, eliminates single control points, and achieves full decentralization.

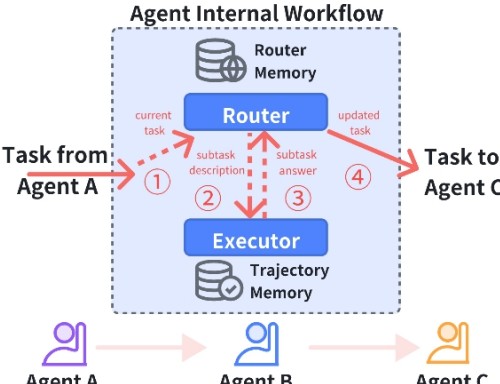

Figure 3: Dual-role agent architecture.

Over tasks, the weight matrix $w_m$ will evolve based on collaborative success, and the edges with a lower weight than a hyper-parameter threshold $\theta_w$ are periodically pruned:

$$\mathcal{E}_{m+1} = \{(a_i^{m+1}, a_j^{m+1}) \mid w_{m+1}(i,j) > \theta_w\}. \tag{3}$$

This pruning mechanism ensures that the network maintains efficient pathways while eliminating unproductive connections, optimizing both communication overhead and routing efficiency.

### 3.3 Adaptive Learning and Specialization

AgentNet's adaptive learning mechanism allows agents to continuously improve and naturally specialize through task experiences, without explicit role assignments. This capability distinguishes AgentNet from static multi-agent systems, enabling it to adapt dynamically to evolving requirements. Agents utilize the *ReAct* (Reasoning + Acting) framework [28, 34], reasoning carefully about task queries and contexts before executing actions. To enhance their reasoning, agents employ Retrieval-Augmented Generation (RAG) [13, 6, 34], retrieving relevant fragments from past experiences stored in memory.

Specifically, each agent $a_i \in \mathcal{A}$ maintains two dedicated memory modules: $\mathbb{M}_i^{\mathrm{rou}}$ for routing and $\mathbb{M}_i^{\mathrm{exe}}$ for execution. These modules store fragments of trajectories corresponding only to steps involving the agent itself, rather than full task trajectories involving all agents. For each memory type $r \in \{\mathrm{rou}, \mathrm{exe}\}$, each memory fragment $f^r = (o^r, c^r, a^r)$ consists of an observation $o^r$ (task query), context $c^r$ (partial task history), and action $a^r$ (agent's response). Upon receiving a new task $t_{m+1}$, the agent retrieves the $k$ most relevant fragments from each memory module, formally defined as:

$$\mathrm{Select}(\mathbb{M}_i^r, t_{m+1}, k) = \arg\max_{\substack{\mathbb{F} \subset \mathbb{M}_i^r \\ |\mathbb{F}|=k}} \sum_{f \in \mathbb{F}} \mathrm{sim}\left(\mathrm{embed}(o_f^r, c_f^r), \ \mathrm{embed}(o_{t_{m+1}}^r, c_{t_{m+1}}^r)\right) \tag{4}$$

Here, $\mathrm{embed}(\cdot)$ is a semantic embedding function that projects the input context into a high-dimensional vector space, and the fragments with the highest relevance are retrieved to inform the agent's reasoning or action for both routing and execution processes.

Both the reasoning and acting processes are enhanced by the retrieval of historical task fragments, allowing the agent to make better decisions based on prior experiences. The reasoning function for each module type is modeled as:

$$\mathcal{R}_{a_i}(t_{m+1}, r) = \mathcal{F}_{\mathrm{reason}}(o_{t_{m+1}}, c_{t_{m+1}}, \{f_j^r\}_{j=1}^k), \tag{5}$$

where $\mathcal{F}_{\mathrm{reason}}$ represents the large language model that serves as the backbone of the LLM Agent, processing the inputs to generate reasoned decisions. The reasoning function $\mathcal{R}_{a_i}$ takes the current observation and question $o_{t_{m+1}}$, the historical context $c_{t_{m+1}}$ representing the partial task trajectory and interactions up to the current point, and the retrieved fragments $\{f_j^r\}_{j=1}^k$ as input to generate the reasoning output. The fragments allow the agent to reason based on prior experiences that are most relevant to the current situation.

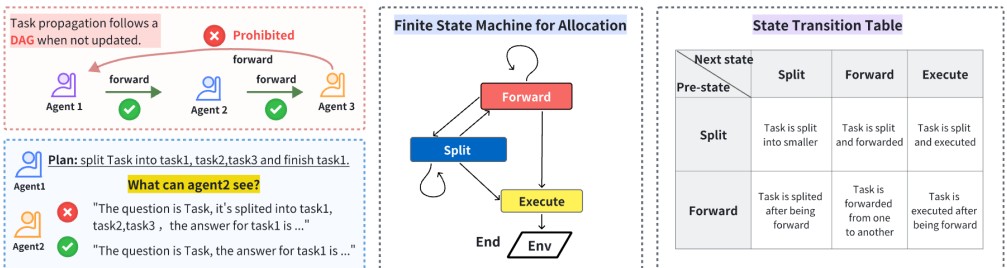

Figure 4: Details of Dynamic Task Allocation.

Once the reasoning process has been completed, the agent executes the chosen action. The action is informed by the reasoning output, which can be expressed as:

$$\mathcal{A}_{a_i}(t_{m+1}, r) = \mathcal{F}_{\text{act}}(o_{t_{m+1}}, \mathsf{c}_{t_{m+1}}, \mathcal{R}_{a_i}(t_{m+1}, r), \{f_j^r\}_{j=1}^k), \tag{6}$$

where $\mathcal{F}_{\text{act}}$ represents the large language model that serves as the backbone of the LLM Agent, translating reasoning into concrete operations. The $\mathcal{A}_{a_i}(t_{m+1}, r)$ function utilizes the reasoning output $\mathcal{R}_{a_i}(t_{m+1}, r)$ along with the retrieved memory fragments to determine the appropriate action. The specific action depends on the module type: for $r = rou$, the router module may produce actions such as forwarding the task to another agent or splitting it into subtasks; for $r = exe$, the executor module generates a single-step operation or response to directly address the final answer.

Agents employ a dynamic memory management strategy, evaluating stored trajectories based on task context, historical usage, frequency, recency, and uniqueness. When a memory module reaches capacity ($C_{\max}$), agents remove the least useful trajectories to maintain a high-quality memory pool through a prompt-based reasoning process. This adaptive process enables agents to naturally specialize over time, optimizing performance across diverse tasks.

## 3.4 Dynamic Task Allocation

The dynamic task allocation mechanism in AgentNet enables efficient distribution of tasks without centralized coordination, creating a responsive system that optimizes both performance and load balancing. Each task $t \in T$ is formally represented as a tuple $t = (o_t, c_t, p_t)$, where $o_t$ contains the task description in natural language, $c_t$ is a vector of capability requirements, and $p_t$ denotes the priority level. To efficiently process a new task $t_{m+1}$ after completing task $t_m$, AgentNet employs a sophisticated mechanism to select the most suitable initial agent. Agent capability representation and matching form the foundation of task allocation. Each agent $a_i^m$, after completing task $t_m$, possesses a capability vector $cv_i^m$ that is dynamically updated through task performance during system operation. In the initial allocation phase, the system selects an entry agent for $t_{m+1}$ using the following formula:

$$a_{initial} = \underset{a_i \in \mathcal{A}_m}{\operatorname{argmax}} \{\operatorname{sim}(c_{t_{m+1}}, cv_i^m)\}, \tag{7}$$

where $c_{t_{m+1}} = \Phi(o_{t_{m+1}})$ represents the capability requirements of task $t_{m+1}$, $cv_i^m$ denotes the capability vector of agent $a_i$, and $\operatorname{sim}(\cdot, \cdot)$ is a similarity function measuring the match between task requirements and agent capabilities. The capability requirements are determined through different methodologies depending on task complexity:

$$c_{t_{m+1}} = \begin{cases} \Phi_{atomic}(t_{m+1}), & \text{for atomic tasks} \\ \Phi_{compound}(t_{m+1}), & \text{for compound tasks.} \end{cases} \tag{8}$$

For atomic tasks, the system uses a function $\Phi_{atomic}$ that maps task properties to capability requirements based on predefined heuristics. For compound tasks, the function $\Phi_{compound}$ utilizes an instruction set with carefully designed prompts to guide the large language model in analyzing task descriptions and inferring the required capability vectors. Agents are then ranked by their capability matching scores, and the highest-scoring agent is selected as the initial executor.

Once a task is assigned to the initial agent, the agent determines how to process it based on the reasoning results from its router module $rou_i$. As shown in Figure 4, the agent can perform one of three operations:

1. **Forward** ($\mathcal{O}_{\text{fwd}}$): Transfer the task unchanged to another more suitable agent, maintaining the task's original state and preserving the Directed Acyclic Graph (DAG) property of the

Table 1: Performance comparison of different methods across various tasks. In all multi-agent methods, we set **3 agents** for each method to ensure a fair comparison. The best result is in bold, while the second is underlined.

| Backbone | Category | Method | MATH(Acc/%) | BBH(Acc/%) | API-Bank(Acc/%) |
|---|---|---|---|---|---|
| DeepSeek-V3 | Single Agent | Direct | 47.86 | 69.00 | 26.00 |
| | | React | 77.14 | 88.00 | 29.00 |
| | | Synapse | 89.28 | 92.00 | 28.00 |
| | | Self-Consistency | 88.00 | 85.00 | 29.00 |
| | | Self-Refinement | 87.14 | 84.00 | 25.00 |
| | Multi-Agent | MorghAgent | 39.29 | 56.00 | 16.00 |
| | | MetaGPT | 92.14 | 64.00 | 22.00 |
| | | AFLOW | 91.67 | 88.00 | 28.00 |
| | | GPTSwarm | 72.14 | 90.00 | 21.00 |
| | | **AgentNet** | **92.86** | **94.00** | **30.00** |
| GPT-4o-mini | Single Agent | Direct | 31.43 | 59.00 | 15.00 |
| | | React | 55.71 | 80.00 | 24.00 |
| | | Synapse | 77.14 | 79.00 | 22.00 |
| | | Self-Consistency | 54.28 | 85.00 | 22.00 |
| | | Self-Refinement | 68.57 | 81.00 | 23.00 |
| | Multi-Agent | MorghAgent | 80.71 | 56.00 | 16.00 |
| | | MetaGPT | 73.57 | 53.00 | 19.00 |
| | | AFLOW | **85.00** | 75.00 | 21.00 |
| | | GPTSwarm | **85.00** | **86.00** | 13.00 |
| | | **AgentNet** | **85.00** | **86.00** | **29.00** |
| Qwen-turbo | Single Agent | Direct | 37.85 | 57.00 | 27.00 |
| | | React | 53.57 | 69.00 | 23.00 |
| | | Synapse | 67.14 | 68.00 | 24.00 |
| | | Self-Consistency | 64.28 | 70.00 | 28.00 |
| | | Self-Refinement | 76.43 | 74.00 | 23.00 |
| | Multi-Agent | MorghAgent | 16.43 | 56.00 | 9.00 |
| | | MetaGPT | 63.57 | 51.00 | 20.00 |
| | | AFLOW | **82.14** | 57.00 | 22.00 |
| | | GPTSwarm | 79.29 | 75.00 | 30.00 |
| | | **AgentNet** | 81.43 | **92.00** | **32.00** |

routing path. Forwarding decisions are based on analyzing the gap between the current agent's capabilities and the task requirements, as well as evaluating the capability vectors of other agents in the network.

2. **Split** ($\mathcal{O}_{\text{split}}$): Decompose the task into subtasks, execute portions matching the agent's expertise, and route the remaining subtasks to an appropriate agents. Subtask routing follows this formula:
$$a_{next} = \underset{a_k \in \mathcal{A}_m \setminus \{a_i\}}{\arg\max} \{\text{sim}(\Phi(o_{t_{m+1}}), c_k^m)\}, \tag{9}$$

where $\Phi(o_{t_{m+1}})$ represents the capability requirements derived from the observation of subtask $j$, determined through the current agent's task decomposition reasoning, and $\mathcal{A}_m \setminus \{a_i\}$ denotes the set of all agents excluding the current one.

3. **Execute** ($\mathcal{O}_{\text{exec}}$): Complete the entire task without further delegation.

A key design feature in the system is that when an agent chooses to split a task, it only forwards the results of the subtasks it has completed, and not the reasoning behind the decomposition. This prevents the transfer of unnecessary information and ensures that task decomposition errors made by one agent do not propagate to other agents in the network.

The agent capability vector $cv_i^m$ is updated based on task execution history and success rates, using the following formula:
$$c_i^{m+1} = \beta \cdot cv_i^m + (1 - \beta) \cdot \Delta c_i^{m+1}, \tag{10}$$

where $\beta \in [0, 1]$ is a decay factor balancing historical capabilities with newly acquired ones, and $\Delta c_i^{m+1}$ represents the new capability contribution demonstrated by the agent in task $t_{m+1}$, calculated by analyzing the types of operations successfully executed by the agent and the quality of results.

The task state updates only when an agent completes a part—by executing or splitting it. Upon subtask completion, the agent updates the context and passes it to the next agent:
$$\text{context}_{\text{updated}} = \text{context}_{\text{original}} \oplus \text{result}(a_j, t_i). \tag{11}$$

While the task is simply forwarded from one agent to another, its state remains unchanged, preserving the Directed Acyclic Graph (DAG) structure of the routing path. This prevents infinite loops and

ensures effective task progression across agents. Through this dynamic task allocation mechanism, AgentNet adaptively optimizes task flow based on task characteristics and evolving agent capabilities.

## 4 Experiment

### 4.1 Main Results

Table 1 summarizes performance across Math, Coding, Logical QA tasks and API-Calling tasks, detailed settings and implementation details are provided in Section C. For Math, Logical QAs and API-Calling tasks, accuracy is reported; for Coding, we report average test case pass rate and full problem pass ratio. Compared to single-agent methods (e.g., Synapse, ReAct), AgentNet achieves competitive or superior performance across all tasks. While ReAct performs well on Math and Logic, its static prompting strategy limits generalization to more complex tasks. Against multi-agent baselines, AgentNet consistently outperforms centralized frameworks such as MetaGPT, which suffers from limited scalability—e.g., only 53.00% accuracy on Logical QA. MorphAgent underperforms on Coding tasks, as it generates self-constructed test cases during training, resulting in invalid or uncompilable outputs. AgentNet's decentralized coordination and retrieval-augmented memory contribute to its robustness across domains, particularly in tasks requiring contextual understanding and adaptive role specialization.

### 4.2 Experiments on Heterogeneous Agents

To investigate the impact of agent diversity on performance, we designed a heterogeneity experiment across different settings on the BBH task. Agents were tested under four configurations: fully homogeneous (identical models and capabilities), LLM heterogeneity (different language models, same capabilities), skill heterogeneity (same model but varied capabilities), and a combination of both. This design allows us to isolate and analyze how model-level and capability-level diversity influence multi-agent collaboration.

Table 2: BBH Accuracy under Different Heterogeneity Settings (Acc%)

| Setting | Fully Homogeneous | Skill Hetero. | LLM Hetero. | Both Hetero. |
|---------|-------------------|---------------|-------------|--------------|
| 3 Agents | 0.86 | 0.84 | 0.81 | 0.81 |
| 5 Agents | 0.79 | 0.86 | 0.85 | 0.85 |

Results in Table 2 show that the impact of heterogeneity on performance depends on team size. With 3 agents, the fully homogeneous setting performs best, while introducing either model or skill diversity reduces accuracy, suggesting uniform reasoning is more effective in small teams. However, with 5 agents, heterogeneous configurations outperform the homogeneous one, indicating that diversity enhances collaboration and complementary reasoning in larger teams. Overall, heterogeneity may introduce coordination overhead in small groups but offers clear benefits at larger scales.

### 4.3 Ablation Study

**Router Effectiveness in AgentNet**  To evaluate AgentNet's decentralized router, experiments were conducted comparing AgentNet with ablation configurations: "Totally Random", "Random Operations", "Random Next Agent ID" and a centralized "Global Router". Each router manages external routing (selecting the next agent) and internal routing (deciding to forward, split, or execute).

Performance was tested on the BBH task (training: 627 problems, testing: 100 problems), with results in Figure 5. AgentNet outperforms randomized methods, achieving 82.14% accuracy during training and 86.00% during testing. Randomizing operations (forward/split/execute) affects task execution more directly, randomizing next agent ID primarily results in suboptimal task delegation but does not disrupt task completion as severely. These results underscore the critical role of effective routing and suggest that optimizing routing decisions can significantly enhance multi-agent system performance.

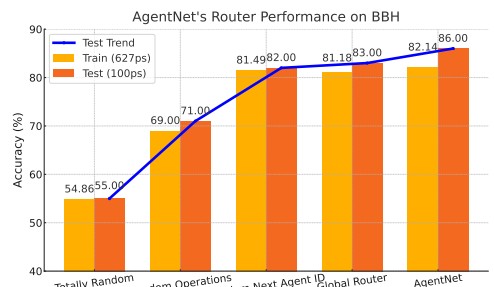

Figure 5: AgentNet's Router Performance on the BBH (Backbone: gpt-4o-mini)

**Impact of Evolution Phase** Results in Table 3 clearly indicate that AgentNet significantly improves performance compared to the non-evolution baseline. On the MATH task, Agent-Net achieves a score of 85.00 versus 77.86. For the function-calling task, performance improves notably from 23.00 to 32.00, and BBH task accuracy rises from 76% to 86%.

Table 3: Performance Comparison of AgentNet vs. Without evolution (Backbone: gpt-4o-mini)

| 3 Agents | MATH | API-Bank | BBH |
|---|---|---|---|
| | (Acc) | Acc | (Acc) |
| w/o evolution | 77.86 | 23.00 | 76.00 |
| AgentNet | 85.00 | 32.00 | 86.00 |

These results confirm that AgentNet's adaptive learning during the evolution phase effectively enhances agent specialization and task performance, demonstrating its essential role in the system's optimization and overall efficiency.

## 4.4 Analysis

**Scalability and Robustness of the System** As shown in Figure 6, both training and testing performance of AgentNet improve slightly with an increased number of agents and a larger executor pool on the BBH task. Training accuracy rose from 80.38 (3 agents, 30 executors) to 81.18 (9 agents, 40 executors), while testing performance ranged from 80 to 86, peaking with 40 executors. These incremental gains highlight AgentNet's ability to scale efficiently through decentralized coordination. While performance improves with more resources, the diminishing returns suggest an optimal configuration exists. Overall, the experiment confirms that AgentNet's adaptive design supports scalable, efficient, and fault-tolerant multi-agent systems.

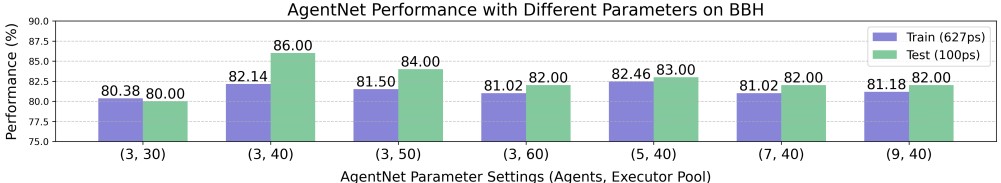

Figure 6: AgentNet Performance with Different Net Parameters. Experiments were conducted with routers without pool limit, where (A, B) represents A as the number of agents and B as the upper limit of the executor pool, with performance evaluated on the BBH.

**Evolution of Agents Networks** The evolution of the agent network in our experiment is illustrated in Figure 7, which demonstrates the transition of a multi-agent system composed of 5 agents running on the BBH (627 pieces) benchmark. The figure captures the network at three key stages: the initial state, an intermediate state, and the final evolved state.

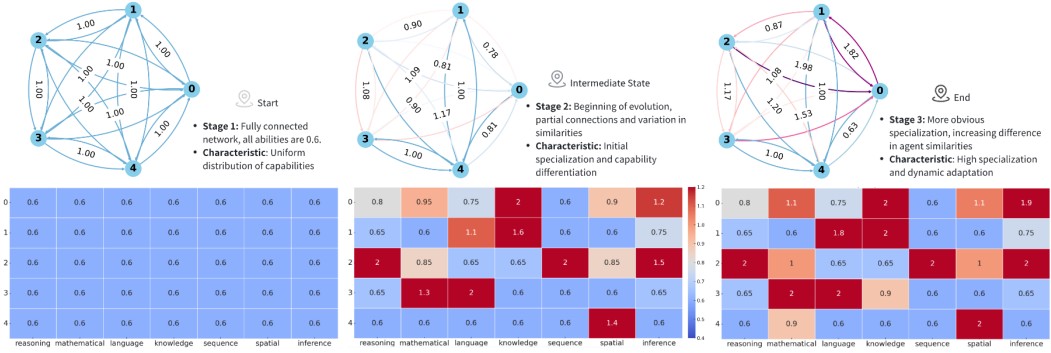

Figure 7: Evolution Example of Agents Networks.

In the initial state, the network is fully connected with uniform connection values of 1.00, indicating equal capabilities among all agents. At this stage, there is no specialization, and all agents are equally equipped to handle tasks. As the network evolves, agents begin to specialize, and connection values vary, reflecting the strength of collaboration. Stronger connections indicate tighter cooperation, while weaker ones suggest less interaction. This evolution shows how agents naturally adapt and form more efficient collaboration patterns. By the final stage, the network exhibits clear specialization, with agents taking on distinct roles. The connection values further emphasize the growing cooperation between specialized agents, improving task performance. This progression demonstrates the

effectiveness of decentralized coordination, where evolving collaboration enhances task allocation, scalability, and fault tolerance.

**Autonomous Specialization of Agents** Based on the observed results in Figure 8, the experiments demonstrate that AgentNet's multi-agent system can naturally specialize agents in a decentralized environment.

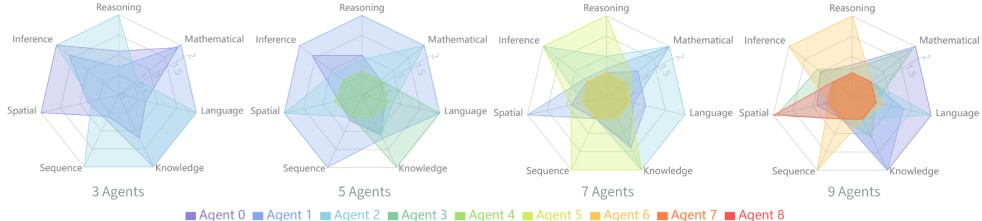

Figure 8: Autonomous Specialization under Different Agent Sets. The upper limit of the executor pool is fixed at 40, regardless of the number of agents. Figure shows agents' fabilities after training on 627 questions on BBH.

With varying numbers of agents and a fixed executor pool of 40 pieces, the ability scores across different tasks such as reasoning, language, knowledge, and sequence showed significant variation. As the number of agents increased, specialization became more evident, particularly in complex tasks, with certain agents excelling in specific areas while others focused on different abilities. This highlights AgentNet's capacity to dynamically refine agent expertise and optimize performance in a decentralized, task-driven system.

**Case Studies** This case study is presented to illustrate the differences between the two methods, based on results obtained using GPT-4o-mini on the BBH dataset. The left image shows the trajectory produced by the ReAct method, while the right image illustrates the trajectory generated by AgentNet. In the case of ReAct, the lack of collective reasoning results in an incorrect response after a single-step inference, highlighting the limitations of the method in handling tasks that require more complex reasoning. In contrast, AgentNet uses a multi-step workflow where agents without the necessary expertise are bypassed, while those with the relevant skills divide the task into smaller steps, leading to a more accurate final solution.

**Question:**
Which sentence has the correct adjective order?
**Options:**
**A.** old-fashioned circular green cardboard exercise computer
**B.** old-fashioned green exercise circular cardboard computer
**ReAct**
**Reasoning:** (B) was selected because it follows a common pattern in English, where opinion adjectives (e.g., "old-fashioned") precede color adjectives (e.g., "green") and shape adjectives (e.g., "circular").

**Action:** (B)                              **Ground Truth:(A)**

Figure 9: ReAct Response with Reasoning

**AgentNet (5 Agents)**
**Agent 0:** Decision: Forward Next Agent: Agent 1
**Agent 1:** Decision: Forward Next Agent: Agent 2
**Agent 2:** Decision: Forward Next Agent: Agent 3
**Agent 3:** Decision: Split
Executable: Adjective order analysis
Delegate: Final answer to another agent
Findings:
   Correct order: Quantity, opinion, size, age, shape, color, origin, material, purpose   Correct Option: (A)
Next Agent: Agent 4
**Agent 4:** Decision: Execute Action: Confirmed option (A)
**Final Outcome:** (A)                    **Ground Truth:(A)**

Figure 10: AgentNet Task Breakdown

## 5   Conclusion

In conclusion, AgentNet provides an effective approach to addressing the limitations of traditional centralized multi-agent systems. With its decentralized architecture, dynamic task allocation, and adaptive learning mechanisms, AgentNet improves scalability, fault tolerance, and task efficiency in collaborative environments. Its privacy-preserving features further ensure secure cooperation across organizations. Our experimental results highlight the advantages of this approach, demonstrating improvements in task efficiency, adaptability, and specialization. AgentNet offers a practical framework for developing more flexible and secure multi-agent systems in dynamic, real-world settings.

# 6 Acknowledgment

This research was supported by Bytedance through a sponsored project that facilitated the execution and completion of this work. The Shanghai Jiao Tong University team is partially supported by National Natural Science Foundation of China (62322603).

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

# A Algorithm of AgentNet

## A.1 Pseudocode of AgentNet

The AgentNet System is designed to optimize task allocation and agent coordination within a multi-agent environment. This algorithm orchestrates the interaction between tasks and agents, aiming to efficiently distribute tasks based on agent capabilities and historical performance.

---

**Algorithm 1** AgentNet System

---

**Require:** Task set $T = \{t_1, t_2, \ldots, t_M\}$
**Ensure:** Optimized network $\mathcal{G}^* = (\mathcal{A}^*, \mathcal{E}^*)$
1: Initialize $\mathcal{A} = \{a_1, a_2, \ldots, a_n\}$, $\mathcal{E}$, $\mathcal{C} = \{c_1, c_2, \ldots, c_n\}$, $w_0$, $\mathbb{M}_i^{rou}$ and $\mathbb{M}_i^{exe}$ $\forall a_i \in \mathcal{A}$
2: **for** each task $t_{m+1} \in T$ **do**
3:     // 1. Task allocation and processing
4:     $c_{t_{m+1}} \leftarrow \Phi(o_{t_{m+1}})$, $a_{curr} \leftarrow \arg\max_{a_i \in \mathcal{A}_m} \text{sim}(c_{t_{m+1}}, cv_i^m)$
5:     $task\_state \leftarrow (o_{t_{m+1}}, \emptyset, p_{t_{m+1}})$, $visited \leftarrow \emptyset$, $finished \leftarrow false$
6:     **while** not $finished$ and $a_{curr} \notin visited$ **do**
7:         $visited \leftarrow visited \cup \{a_{curr}\}$
8:         $fragments^{rou} \leftarrow \text{Select}(\mathbb{M}_{curr}^{rou}, t_{m+1}, k)$
9:         $action \leftarrow \mathcal{F}_{act}(o_{t_{m+1}}, c_{t_{m+1}}, \mathcal{F}_{reason}(o_{t_{m+1}}, c_{t_{m+1}}, fragments^{rou}), fragments^{rou})$
10:         **if** $action = \mathcal{O}_{fwd}$ **then**
11:             $a_{curr} \leftarrow \arg\max_{a_k \in \mathcal{A}_m \setminus \{a_{curr}\}} \text{sim}(c_{t_{m+1}}, c_k^m)$
12:         **else if** $action = \mathcal{O}_{split}$ **then**
13:             $subtasks \leftarrow \text{DecomposeTask}(t_{m+1})$
14:             $task\_state.context \leftarrow task\_state.context \oplus$ ProcessSubtasks$(subtasks, a_{curr}, \mathcal{A}_m)$
15:             $finished \leftarrow \text{AllSubtasksCompleted}(subtasks)$
16:         **else**
17:             $task\_state.context \leftarrow task\_state.context \oplus$ ExecuteTask$(a_{curr}, t_{m+1}, \text{Select}(\mathbb{M}_{curr}^{exe}, t_{m+1}, k))$
18:             $finished \leftarrow true$
19:         **end if**
20:     **end while**
21:     // 2. Network update
22:     **for** each interacting pair $(a_i, a_j)$ **do**
23:         $w_{m+1}(i,j) \leftarrow \alpha \cdot w_m(i,j) + (1 - \alpha) \cdot S(a_i^{m+1}, a_j^{m+1}, t_{m+1})$
24:     **end for**
25:     $\mathcal{E}_{m+1} \leftarrow \{(a_i^{m+1}, a_j^{m+1}) \mid w_{m+1}(i,j) > \theta_w\}$
26:     // 3. Agent capability and memory update
27:     **for** each participating agent $a_i$ **do**
28:         $c_i^{m+1} \leftarrow \beta \cdot cv_i^m + (1 - \beta) \cdot \Delta c_i^{m+1}$, Update $\mathbb{M}_i^{rou}$ and $\mathbb{M}_i^{exe}$
29:     **end for**
30: **end for**
31: **return** $\mathcal{G}^* = (\mathcal{A}, \mathcal{E})$

---

This algorithm provides a systematic approach to managing agent-task interactions, enhancing coordination efficiency, and supporting adaptive learning and specialization within complex environments.

## A.2 Scalability Analysis of AgentNet

We analyze the scalability and computational efficiency of **AgentNet** when deployed with a large number of agents. As shown in Figure 6, AgentNet maintains high and stable performance as the scale of the multi-agent network increases, demonstrating its strong scalability and robustness under network expansion.

After each round of task execution, AgentNet updates its parameters through several lightweight operations. Assuming there are $k$ agents and each agent's memory contains $m$ entries (while ignoring asynchronous parallelism), the computational costs are as follows:

- **Agent memory updates:** Updating each agent's memory requires $O(m)$ time, leading to a total cost of $O(km)$.

- **Capability vector updates:** Each agent's capability vector update takes $O(1)$ time, resulting in a total cost of $O(k)$.

- **Topology updates:** If a task execution trajectory involves $t$ agents (i.e., $(t-1)$ edges), only the communication edges among these $t$ agents need to be updated, which costs $O(t)$, and typically $t \ll k$.

- **Communication overhead:** Each agent only communicates with directly connected agents, avoiding expensive global broadcast operations.

Taken together, the overall time complexity of AgentNet's memory and parameter updates is $\mathbf{O(km)}$ in large-scale applications, which highlights its high scalability.

Moreover, thanks to its fully decentralized design, each agent in AgentNet can be deployed on separate devices, enabling asynchronous execution of subtasks and independent updates of memory and capability vectors. This design substantially reduces both time and bandwidth overhead in large-scale distributed environments.

## B  Supplementary Review of Related Literature

### B.1  LLM-based Multi-Agent Systems

The development of LLM-based multi-agent systems (LaMAS) [27] has advanced rapidly in recent years. Early frameworks, such as AutoGen [25] and MetaGPT [9], made significant strides in establishing foundational architectures for orchestrating multiple LLM agents through structured workflows. AutoGen provided a flexible framework for defining agent interactions, while MetaGPT incorporated software development principles to enhance collaboration. These centralized frameworks proved effective for managing multi-agent interactions. However, they also faced inherent challenges, including limited scalability, single points of failure, and difficulty in dynamically adapting to evolving tasks or incorporating new expertise.

In response to these limitations, more recent frameworks such as AgentScope [5] and MegaAgent [22] have focused on improving robustness and scalability. AgentScope introduced modular design patterns to enhance system reliability, while MegaAgent employed hierarchical structures to scale agent interactions. Although these frameworks offer improvements, they still operate under centralized control paradigms, with a master agent delegating tasks, which continues to lead to scalability bottlenecks and single points of failure. Moreover, existing LaMAS implementations predominantly utilize single-source LLMs, lacking the integration of heterogeneous models. Their workflows are typically static, unable to dynamically allocate resources based on task complexity, further constraining adaptability.

In contrast, AgentNet introduces a novel decentralized approach, addressing these challenges by enabling agents to autonomously refine their expertise and dynamically allocate resources. AgentNet supports scalable, fault-tolerant collaboration without reliance on a central orchestrator, overcoming the limitations of centralized frameworks.

### B.2  Evolutionary Agent Systems

Inspired by natural evolution, recent researchers have explored evolutionary approaches to automate and optimize agent behaviors and workflows in LaMAS. Existing efforts can be broadly categorized into the following areas:

- **Prompt Evolution and Optimization** – Techniques such as PromptBreeder [4], DsPy [12] and AgentPrune [30] apply evolutionary algorithms to iteratively refine prompt generation, improving task performance through better input design.

- **Inter-Agent Topology Optimization** – Systems like GPTSwarm [35], DyLAN [15], and G-Designer [31] focus on evolving the structural organization of agent interactions. These works aim to optimize communication patterns, task allocation, and collaboration efficiency within multi-agent networks.

- **Agent Role and Persona Specialization** – Frameworks such as AgentVerse and MorphAgent [3, 16] refine agent roles and profiles, enabling more effective specialization and coordination among agents in complex tasks.

While these evolutionary approaches have shown promise, they primarily focus on individual agent adaptation rather than collective coordination. Additionally, they still tend to operate within centralized control structures, which limits their scalability and dynamic adaptability. Recent frameworks like AgentSquare [18] and AFlow [32] have begun to formalize automated design processes for agentic systems, improving system-level orchestration and workflow automation. Another key direction is self-adaptive agent architectures, where agents adjust their strategies in real-time based on feedback and accumulated experience. For example, EvoMAC [10] combines reinforcement learning with evolutionary algorithms to optimize agent decision-making and policy updates.

However, these approaches are often limited to single-agent adaptation and lack mechanisms for decentralized specialization and coordination across large-scale agent collectives. While EvoMAC and other systems focus on optimizing individual agents, they are not designed for scalable, multi-agent, decentralized collaboration. In contrast, AgentNet integrates evolutionary learning with decentralized control, enabling heterogeneous agents to dynamically evolve their roles, adapt their strategies in real-time, and collaborate flexibly across a large-scale multi-agent system. This integration of evolutionary learning with decentralized control makes AgentNet a more suitable framework for real-time, adaptive, and scalable multi-agent collaboration.

## C  Experimental Setup

**Tasks and Benchmarks**    We evaluate methods using several benchmarks across three task categories, along with custom constructed training and test sets for each benchmark:

- **Mathematics**: This task involves mathematical problem and is evaluated using MATH [8], which includes problems with 7 different types. The training set consists of 100 examples per type (total of 700 problems), while the test set consists of 20 examples per type (total of 140 problems).

- **Logical Question Answering**: This task tests reasoning and logical question answering abilities using the BBH (Big-Bench Hard) benchmark [21]. The training set follows the MorphAgent setup, selecting 627 examples from 20 tasks. For testing, each task has 5 examples of varying difficulty, totaling 100 test problems.

- **Function-Calling**: This benchmark evaluates the agent's ability to perform tool-augmented task planning and API usage, based on the API-Bank dataset [14]. We construct a training set of 100 tasks and a test set of 100 tasks, randomly sampled from the full API-Bank corpus. Since the original dataset does not include category labels, we annotate each task using GPT-4o-mini to assign one of the seven task types: *health*, *account*, *schedule*, *information*, *housework*, *finance*, and *others*. Each task is further categorized into one of three difficulty levels, determined by prompt complexity and required toolchain length.

**Metrics**    A range of evaluation metrics have been adopted for different tasks. For the Mathematics and the Logical Question Answering tasks, the accuracy metric is utilized to evaluate the consistency of the output answer with the true answer within the specified format. For the Coding task, the average test case pass rate (i.e., the ratio of the number of passed test cases to the total number of test cases) and the ratio of problems passed across all test cases have been employed as the evaluation metrics.

**Baselines**    We compare AgentNet with two categories of baselines: single-agent and multi-agent frameworks:

- **Single-agent frameworks:** These methods involve a single agent solving tasks independently without collaboration or coordination with other agents.

    - **Direct**: A baseline approach where the LLM directly generates outputs.
    - **Chain of Thought**: A prompting technique that elicits step-by-step reasoning from language models [24].

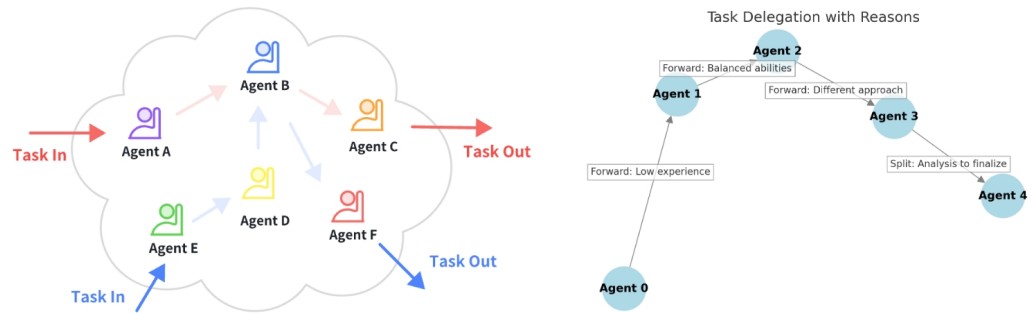

Figure 11: Illustration of the execution process (passing and forwarding) of tasks in AgentNet

- **Synapse**: A trajectory-as-exemplar prompting method, which prompts the LLM with complete trajectories of the abstracted states and actions to improve multi-step decision-making. [33]
- **Self-Consistency**: A decoding strategy that samples multiple reasoning paths and selects the most consistent answer through majority voting, enhancing reliability [23].
- **Self-Refinement**: An iterative approach where models critically evaluate and improve their own solutions over multiple passes, progressively enhancing solution quality [17].

- **Multi-agent frameworks:** These methods involve multiple agents working collaboratively to solve tasks, each contributing to different aspects of the task-solving process.

- **MetaGPT**: A software development framework where specialized agents (like product manager, architect, engineer) collaborate in a waterfall workflow to complete complex engineering tasks [9].
- **AFLOW**: A framework that optimizes agent workflows using Monte Carlo Tree Search over code-represented workflows with execution feedback [32].
- **GPTSwarm**: A framework modeling agents as computational graphs with automatic optimization of both prompts and agent collaboration patterns [35].
- **MorphAgent**: A framework featuring self- evolving agent profiles that dynamically optimize individual expertise in the profile through three metrics [16].

**Parameter Configuration** In our implementation, we configure the LLM API with a temperature of 0.0, a maximum token limit of 2048, and a top-p value of 1.0, ensuring consistent results throughout our experiments and enabling reliable comparisons and analysis. For the memory pool experiment, we utilize the "BAAI/bge-large-en-v1.5" model to compute the similarity between task queries and database trajectories.

## D   AgentNet Experiment Configuration

### D.1   Task to Ability Mapping in AgentNet

This table illustrates the mapping between a diverse set of cognitive tasks and their corresponding core cognitive abilities. Each task is linked to one or more foundational abilities—such as reasoning, inference, mathematical competence, or linguistic understanding—which are represented using distinct color encodings to enhance visual clarity and interpretability.

| Task in BBH | Abilities |
|---|---|
| boolean_expressions | reasoning |
| logical_deduction_three_objects | reasoning |
| logical_deduction_five_objects | reasoning |
| logical_deduction_seven_objects | reasoning |

| Task | Abilities |
|------|-----------|
| causal_judgement | reasoning, inference |
| formal_fallacies | reasoning, inference |
| tracking_shuffled_objects_three_objects | reasoning, sequence |
| tracking_shuffled_objects_five_objects | reasoning, sequence |
| tracking_shuffled_objects_seven_objects | reasoning, sequence |
| multistep_arithmetic_two | mathematical |
| geometric_shapes | mathematical, spatial |
| object_counting | mathematical, spatial |
| word_sorting | mathematical |
| date_understanding | mathematical, language |
| dyck_languages | mathematical, language |
| disambiguation_qa | language |
| hyperbaton | language |
| salient_translation_error_detection | language |
| movie_recommendation | knowledge |
| sports_understanding | knowledge |
| penguins_in_a_table | knowledge |
| reasoning_about_colored_objects | knowledge |
| ruin_names | language, knowledge |
| temporal_sequences | sequence |
| navigate | spatial |
| web_of_lies | inference |
| snarks | inference |

The construction of this task-ability mapping is motivated by the objective of optimizing agent-task alignment within a multi-agent system. By adopting a single-level ability classification, we establish a tractable yet expressive abstraction of agent competencies that facilitates efficient task routing and capability differentiation.

This initial flat taxonomy serves as a foundational layer for scalable skill orchestration. It provides a principled basis for agent coordination and lays the groundwork for future extensions toward a hierarchical ability framework—enabling finer-grained specialization and more precise agent-task matching in complex environments.

## D.2 Initial Configuratio

The initial configuration specifies the setup for each experiment conducted within the AgentNet framework. This YAML-based configuration file defines global experimental parameters, default agent initialization, and the structure of the agent communication network.

```
experiment_config:
 LLM: gpt-4o-mini
 dataset: bigbenchhard
 task_num: 100
 agent_num: 3
 forward_path_max_length: 3
 max_execution_times: 5 ## for tuning
 user_react: True
```

```
default_agent_config: &default_agent_config
 LLM: gpt-4o-mini
 abilities:
  reasoning: 0.6
  mathematical: 0.6
  language: 0.6
  knowledge: 0.6
  sequence: 0.6
  spatial: 0.6
  inference: 0.6
 executor_memory_limit: 40
 embedding_cache_limit: 1000
 router_memory_limit: -1
 decay_rate: 0.1
 decay_interval: 10
 router_retrieval_num: 3
 executor_retrieval_num: 3

agent_graph_config:
 graph_type: complete
```

Listing 1: Experiment Initial Configuration YAML

The configuration includes three primary components:

- **Global Experiment Settings (`experiment_config`)** This section controls the overall experiment, including:

  - The underlying LLM used by agents (`LLM`).
  - The evaluation dataset (`dataset`).
  - Number of tasks to be dispatched (`task_num`) and number of agents (`agent_num`).
  - Routing and execution limits (`forward_path_max_length`, `max_execution_times`).
  - Whether the React output format (`user_react`) is enabled, as it allows for reasoning before taking action.

- **Default Agent Parameters (`default_agent_config`)** All agents are initialized with this shared configuration unless overridden. It includes:

  - A uniform initialization across all ability dimensions (e.g., `reasoning`, `language`, `spatial`, etc.).
  - Memory and caching limits (`executor_memory_limit`, `embedding_cache_limit`).
  - Temporal dynamics of memory decay (`decay_rate`, `decay_interval`).
  - Retrieval settings for routing and execution modules.

- **Agent Communication Graph (`agent_graph_config`)** Defines the topology of inter-agent communication. In this example, a *complete graph* is used, enabling each agent to communicate with all others.

### D.3 AgentNet Prompt Specification

The Agent Prompt Specification defines the structured input schema received by each agent in the AgentNet system. This schema ensures that all relevant contextual, historical, and task-specific information is explicitly provided before an agent begins reasoning or execution.

Rather than free-form prompts, AgentNet adopts a structured prompt interface, where each field conveys a distinct aspect of the task environment. This design improves interpretability, modularity, and control, making it easier to route tasks, compare decisions, and analyze agent behavior.

Table 5: Schema definition for AgentNet prompting.

| Field | Description |
|---|---|
| `major_problem` | The overarching goal or task shared among agents in the system. |
| `experiences` | Past task examples or performance records relevant to the current task. |
| `task_context` | Previously completed subtasks and results, used to determine next steps. |
| `current_agent_info` | Information about the current agent, including its abilities and status. |
| `task_type` | The category or nature of the task (e.g., reasoning, language, etc.). |
| `task_description` | A detailed description of the specific problem to be addressed. |
| `agent_info` | Information about other agents in the system, including their abilities and prior performances. |
| `constraints` | Format or structural requirements for the expected result output. |
| `thought` | Reasoning or intermediate planning used to guide execution of the task. |

This input schema acts as the interface contract between the system and each agent, standardizing how agents perceive their operating context. It also lays the foundation for future work on agent generalization, multi-agent coordination, and prompt optimization.

# E  Limitations and Future Work

Despite AgentNet implementing a fully distributed, adaptive learning multi-agent system (MAS) with dynamic task allocation, several important limitations remain that require further exploration in future work.

One key challenge is how to improve task performance in heterogeneous agent environments. In real-world applications, agents often vary significantly in terms of model capabilities, workflow structures, tools, and available data. The impact of such heterogeneity on AgentNet's performance, especially in terms of task coordination and resource allocation, remains an open question. Understanding how to adapt the system to handle such variations efficiently will be crucial for its scalability and effectiveness in complex environments.

Secondly, the decision-making process of the router within each agent, particularly in relation to exploration and discovery, requires more in-depth study. Currently, the router selects agents from a relatively small pool of predefined candidates. However, in larger-scale systems involving hundreds or potentially thousands of agents, the challenge of accurately identifying the most suitable agent for task delegation becomes significantly more complex. This problem is further compounded in heterogeneous settings, where different agents may possess distinct strengths and weaknesses. To address this, future research could focus on developing more sophisticated routing mechanisms that can autonomously identify and delegate tasks to the most appropriate agents, even in large and diverse agent pools.

Additionally, a promising direction for future work involves designing incentives that encourage the router to explore agents beyond the predefined candidate set. By enabling AgentNet to dynamically discover new agents or specialized capabilities, such an approach would enhance its adaptability and scalability, ultimately improving the system's overall performance and autonomy.

## F Ethics Statement

This study did not involve human participants, animal subjects, or the use of personal data. All datasets and benchmarks employed are publicly available and used in accordance with their respective licenses. Therefore, no ethics approval was required.

