# OpenReview forum: "AgentNet: Decentralized Evolutionary Coordination for LLM-based Multi-Agent Systems"
_NeurIPS.cc/2025/Conference — NeurIPS 2025 poster_

### Official Review · Reviewer_P2gV · 2025-06-09

**Clarity:** 2
**Significance:** 2
**Originality:** 2
**Rating:** 3
**Confidence:** 3

**Summary:**

The authors propose a new multi-agent system (MAS) with 1) decentralized agent coordination, 2) a dynamic agent communication and cooperation pattern, and 3) an agent skill refinement mechanism based on a retrieval-based memory system. Some experimental results show that the proposed MAS can outperform many existing single agents and MAS on different tasks, and other ablation studies show that the design of routing and evolution are essential to the whole system.

**Questions:**

Please refer to the Weaknesses above. Some other additional questions are as follows:
1. As shown in the Appendix E case studies, does it mean the task always starts with Agent 0?
2. What if agents are coming to different answers to a specific step? How does the multi-agent framework handle that inconsistency or hallucination? Will the latter one always rewrite the earlier one?

**Ethical Concerns:**

["NO or VERY MINOR ethics concerns only"]

**Final Justification:**

I will maintain my score after going through the response and other reviews. The proposed mechanism seems to lack a mechanism to prevent or handle the cyclic dependency, and the mechanism heavily relies on careful prompting.

**Limitations:**

Yes

**Quality:**

2

**Strengths And Weaknesses:**

Strengths:
1) The design of the new MAS is flexible and requires less human prior knowledge when building a system for specific tasks.
2) The experimental results seem promising and outperform many existing (multi-)agent mechanisms.


Weaknesses:
1) The system's design may have limitations and may require additional clarification on its generality.
  As the proposed MAS focuses on the decentralized setting with DAG design, it is unclear how the system can handle cases requiring cyclic dependencies. For example, if a task consists of subtask A, B, and C, and subtask A requires special domain knowledge of Agent 1, solving subtask B requires the result of subtask A and special domain knowledge of Agent 2, and solving subtask C requires the result of subtask B and special domain knowledge of Agent 1.
  - It is unclear from the paper whether it is possible to construct a warm-up dataset (referred to as a training set in the appendix) that can help different tasks or whether each special kind of task requires a special dataset to warm up the system again.

2) The presentation of the paper can be improved.
  - It is unclear what is the relation between the context ($c_f$ or $c_{t_{m+1}}$) and the agent capability vector $c_i^m$.
  - It is unclear how to find the initial agent. Although Section 3.4 tries to discuss the mechanism, it is still confusing even for a basic understanding of whether such a mechanism is an independent module or reusing the forward/split function of a special agent.
  - Too many vague words are used to describe the mechanism but give no information, e.g., "a sophisticated mechanism" and "carefully designed prompts."

3) It is unclear how the capabilities/abilities of the agent are initialized.
  - The definitions of "capability" and "abilities" are difficult for readers to distinguish.
  - It is unclear whether the experiments cover the scenario of a different (disjoint) set of capabilities/abilities.

---

> ### Author Rebuttal · Authors · 2025-07-30
>
> >**Q1.Cyclic Dependencies**
>
> **A1**
> We sincerely thank the reviewer for the thoughtful analysis of our work. We would like to further clarify the generality of AgentNet in the scenario you described:
>
> In the case mentioned, **AgentNet will not fall into a cycle**. Specifically, Agent 1 would first analyze the entire task t and determine that subtask **A** can be completed by itself, while subtasks **B** and **C** require the involvement of other agents. Agent 1 will then isolate and complete subtask **A** and forward the query and result of subtask **A**, together with the original task t, to Agent 2 through the router.
>
> Upon receiving the original task t and the query and result of subtask **A**, **Agent 2** will leverage its own knowledge base to isolate and complete subtask **B**. It will then forward the combined information in the same manner.
>
> Finally, when **Agent 3** completes subtask **C**, it will determine that the task has reached its termination condition and output the final result.
>
> This sequential execution and information passing design ensures that AgentNet can handle tasks with dependent subtasks without falling into cyclic dependencies.
>
> Additionally, as described in **Section 3.4: Dynamic Task Allocation**, we have implemented a **directed acyclic graph (DAG)** structure for task passing. In this structure, once a task has been forwarded to an agent, it will not be forwarded back to any agent that has already processed that task or received the information. This ensures that the task propagation avoids any cycles, and that deadlock situations are prevented. In other words, the system guarantees that each task progresses in a one-way flow, which prevents the system from getting stuck in cyclic dependencies or infinite loops.
>
> We hope this clarification addresses the concerns raised, and we appreciate the reviewer's valuable feedback.
>
>
> ---
>
>
> >**Q2.Dataset Construction**
>
> **A2**
> In the experiments presented in this paper, to maintain consistency, the warm-up dataset and the validation dataset were obtained by randomly splitting the full dataset.  Therefore, the warm-up dataset contains a uniformly random distribution of task types, which allows it to help the system handle various types of tasks.
>
> Importantly, all methods that require warm-up are trained on the exact same warm-up dataset, ensuring a fair comparison. We also ensure that no task in the warm-up dataset appears in the test set, thereby preventing any data leakage or memorization effects.
>
> In future work, we plan to explore constructing warm-up datasets specifically tailored for different task types and performing warm-up or re-warm-up on AgentNet accordingly.
>
> We appreciate the reviewer for raising this point, and we will further analyze the design of the warm-up dataset in future work.
>
>
>
> ---
>
>
> >**Q3.Presentation of the Paper**
>
> **A3**
>
> We sincerely thank the reviewer for raising these valuable concerns about the clarity of the paper. We provide detailed clarifications as follows:
>
> 1. **Relation between context and capability vector.**
>    The context $c_f$ or $c_{t_{m+1}}$ and the capability vector $c_j^m$ are not directly related. We acknowledge that the low distinguishability of the symbols in the current version of the paper caused this confusion, and we will revise the notation in future versions to improve clarity.
> 2. **Finding the initial agent.**
>    This is an excellent question. In the process of selecting the initial agent, we reuse the **forward/split function of a special agent**. Specifically, we first randomly assign the task to one agent, and then the router of that agent determines the first agent to receive and process the task using the forward/split function.
>    During our experiments, we identified a significant challenge: as the scale of AgentNet grows larger, finding an appropriate initial agent becomes increasingly difficult. We will address this challenge as part of future work.
> 3. **Vague wording in the paper.**
>    We acknowledge that the paper currently contains some unclear expressions (e.g., "a sophisticated mechanism" and "carefully designed prompts"). We will carefully review and revise such wording in future versions to provide more precise descriptions.
>
> We appreciate the reviewer’s feedback and will make the necessary improvements in the revised version.
>
>
>
> ---
>
>
> >**Q4.Initialization of Capabilities/Abilities**
>
> **A4**
>
> We provide the following clarifications:
> 1. The initialization of capabilities/abilities is manually predefined.
> 2. There is actually **no difference** between "capability" and "ability" in our paper;   these terms were used interchangeably to refer to the same concept.   We apologize for the confusion this may have caused and will unify the terminology in future versions.
> 3. Please refer to **Table 2 (Heterogeneity Experiment)**.   The "Skill Hetero." setting in the table represents an AgentNet composed of heterogeneous agents with different predefined capabilities at initialization.   Our findings show that the impact of capability heterogeneity is related to the network size:
>     - In small networks, uniform reasoning (homogeneous capabilities) tends to be more effective.
>     - In larger networks, agents specializing in different capabilities (heterogeneous) collaborate more effectively.
>
> We will revise the text in the updated version to clearly state these points and avoid ambiguity.
>
>
>
> ---
>
> >**As shown in the Appendix E case studies, does it mean the task always starts with Agent 0?**
>
> **A5**
>
> With respect, **not all initial agents are Agent 0**.  However, we did observe an interesting phenomenon: during the very early stages of the warm-up phase, or at certain points in the middle of the process, when an agent faces neighboring agents with **identical capabilities, success rates, and edge weights**, the agent’s LLM tends to forward the task to the agent with the smaller ID number.  In these cases, the reasoning provided by the LLM is often simply a preference for the agent with the smaller ID.
>
> To prevent this type of ID-based bias, the AgentNet codebase is designed such that when the router encounters multiple neighboring agents with equal capabilities, it uses the **random** function to select the next agent. This better reflects the intended decentralized behavior of the system and ensures fairness in agent selection.
>
> This mechanism has shown stable performance in practice and does not negatively affect overall task execution.
>
> ---
>
> >**What if agents are coming to different answers to a specific step? How does the multi-agent framework handle that inconsistency or hallucination? Will the latter one always rewrite the earlier one?**
>
> **A6**
> In the early stages of our research, we also observed issues in the initial version of AgentNet where multiple agents could produce inconsistent answers or even fall into cyclic responses. However, after carefully refining the prompts and adjusting the framework, the frequency of such hallucination phenomena has been significantly reduced.
>
> In fact, AgentNet inherently includes a behavior pattern that helps mitigate conflicting outputs in a multi-agent setting. Specifically, in the early stage of system evolution, we observed that when an agent (e.g., Agent A) generates an answer it deems potentially complete, but lacks sufficient experience or capability confidence, it tends to forward both the intermediate result and the original task to a neighboring agent for validation. This built-in verification behavior serves as a natural check-and-balance mechanism across agents.
>
> As the system continues to evolve and each agent accumulates its own specialized skills and successful execution history, the likelihood of hallucination or “agents misleading one another” is naturally reduced. Agents develop confidence in specific task types where they have historically succeeded, and therefore are less likely to defer to peers unnecessarily, leading to more stable and accurate outputs overall.
>
> We plan to open-source the full code and prompts, which will allow the community to further explore, analyze, and reproduce these behaviors.

---

> > ### Author Response · Authors · 2025-08-04
> >
> > Thank you for your thoughtful review. All of your concerns have been carefully addressed. We look forward to any further discussion！

---

> > > ### Comment · Area_Chair_oW3g · 2025-08-05
> > > **Engage in the author-reviewer discussion**
> > >
> > > Dear Reviewer P2gV,
> > >
> > > Please also engage in discussion with an official comment. Any follow-up questions? Reflections on why the rebuttal changed your opinion on the submission? Or why it did not?

---

> > ### Comment · Reviewer_P2gV · 2025-08-05
> >
> > Thanks for the clarification from the authors. I tend to maintain my score. The key reasons are 1) several writing problems may need to be addressed to produce a clear version of the paper, and 2) the system handles hallucination and inconsistency with prompts, which may not be reliable enough for cases that depend on factual information.

---

> > > ### Author Response · Authors · 2025-08-08
> > >
> > > We sincerely thank the reviewer for engaging with our work and for acknowledging our clarifications.  We would like to offer the following responses to the comments on writing clarity and potential limitations in reliability.
> > >
> > > On the writing issues, we appreciate the suggestion.  As mentioned in our rebuttal (A3), we acknowledge that the current version contains some vague expressions.  We are committed to revising these thoroughly in the next version.  If the reviewer has specific areas or examples that remain unclear after our clarifications, we would be grateful for further guidance.
> > >
> > > Regarding hallucination and inconsistency, our rebuttal (A6) addresses this concern in detail.  We describe AgentNet’s built-in verification mechanism, which helps mitigate conflicting outputs.  We also provide empirical observations showing a reduction in hallucination frequency.  Furthermore, the agents progressively develop confidence in specific tasks through specialization.  To our knowledge, such mechanisms are rarely explored in existing multi-agent system designs.  We believe this represents a meaningful technical contribution.
> > >
> > > We respect the reviewer’s decision to maintain the current score. We believe our clarifications show that the concerns have been effectively addressed.  We would greatly appreciate it if the reviewer could reconsider the evaluation in light of these responses.

---

### Official Review · Reviewer_RQrj · 2025-06-18

**Clarity:** 3
**Significance:** 3
**Originality:** 3
**Rating:** 4
**Confidence:** 2

**Summary:**

This paper introduces a decentralized method designed for MAS grounded in LLMs. This is achieved through a dynamic directed acyclic graph for task routing, fostering enhanced system robustness and emergent intelligence. Furthermore, the framework incorporates a dynamic agent graph topology, where agent nodes and connections are adjusted in real-time based on task requirements and agent capabilities. Finally, a retrieval-augmented memory system is employed to support continuous skill refinement and agent specialization. The experiments are presented on three datasets.

**Questions:**

Please see the above weaknesses.

**Ethical Concerns:**

["NO or VERY MINOR ethics concerns only"]

**Final Justification:**

The proposed method uses a decentralized graph-topology multi-agent system for collaborative task execution and evolution. The authors address the issues regarding the clarity and experimental results of paper.

**Limitations:**

Yes

**Quality:**

3

**Strengths And Weaknesses:**

Strengths：
1. The proposed method uses a decentralized graph-topology multi-agent system for collaborative task execution and evolution. Agents become more specialized and perform distinct roles, which seems meaningful.
2. The experiment is relatively comprehensive and demonstrates the efficiency, adaptability, and scalability of the method in a dynamic environment.

Weaknesses:
1. In Figure 1, the excessive adjectives for multi-agent are confusing. Some terms are unexplained, like 'hierarchical'. There are also same meaning words, such as 'fixed' and 'static'. Although the three keywords presented for Pre-Defined Agents and Self-Evolving Agents are refined key features, they can't form a contrast. For example, 'static' corresponds to 'decentralized', but in the key features, 'static' corresponds to 'dynamic'.
2. The computation of some variables is unclear. In formula 2, it is not explained how the success rate $S$ is calculated. In formula 5, the reasoning function uses the historical context $c_{t_{m+1}}$ as input to generate the reasoning output. What specific historical content is included in $c_{t_{m+1}}$? Additionally, how is the $p_t$ in the task triplet constructed?
3. In Table 1, why does the AFLOW method outperform AgentNet on the Qwen-turbo backbone?
4. Some hyperparameters need further discussion, such as $\alpha$, $\beta$, and $\theta_w$.
5. Figures 1, 2, and 5, as well as Table 2, seem not to be cited in the main paper.
6. On line 5 from the bottom of page 8, there is a missing number at the '-' position.

---

> ### Author Rebuttal · Authors · 2025-07-30
>
> >**Q1.Clarity of Paper**
>
> **A1**
> We thank the reviewer for pointing out areas where the writing of our paper could be improved, which we acknowledge was partly due to the strict page limit.  We agree that certain aspects of Figure 1 require clarification and refinement.  Below, we address the reviewer’s concerns:
>
> **“Hierarchical” vs. “Decentralized.”**
> In traditional centralized multi-agent systems, a hierarchical structure is common: a central coordinator agent sits at the top level and is responsible for planning the tasks to be executed by the system.  This central agent also makes routing decisions, i.e., selects and instructs subordinate agents to perform specific subtasks.
> By contrast, in AgentNet, each agent autonomously makes routing decisions during task execution based on its own knowledge, experience, tools, and information from its neighbors.  There is no central coordinator involved in either task planning or routing decisions.  All agents participate as equals, forming a fully decentralized multi-agent system.
>
> **“Static” vs. “Adaptive.”**
> In traditional centralized systems, role assignments for participating agents (e.g., coordinator, programmer, summarizer) are typically static—roles are fixed and remain unchanged throughout the task execution process.
> In AgentNet, roles are adaptive and flexible.  During long-term, dynamic task execution in real-world settings, each agent continuously updates its experience memory and capability vectors.  These updates enable the agent to evolve its capabilities and adapt the responsibilities it assumes in different tasks, rather than being bound by pre-defined roles.
>
> The reviewer’s  comments regarding the clarity of Figure 1 is valuable. In the revised version, we will adjust the keywords and their placement in Figure 1 to make the contrasts clearer and avoid confusion.
>
>
>
> ---
>
> >**Q2.Clarity of Paper – Variable Computation**
>
> **A2**
>
>
> Thank you so much for raising this concern. We acknowledge that certain parts of the paper could be improved in terms of writing clarity. We would like to clarify the following:
>
> 1. **Success rate in Formula 2.**
>    The success rate S refers to the success rate of communication edges between two agents in the task execution chain after the current task has been completed. Specifically, when a task is finished and its correctness is verified, we compute the success rate for the edge where one agent routes the task to the next agent. This is defined as:
>
> ```
> S = {successful tasks of this type}/{total routed tasks of this type}.
> ```
> 2. **Historical context in Formula 5.**
>    The historical context $c_{t_{m+1}}$ includes all agent observations, planning and reasoning contents, and execution results from all steps **up to the current step**. This context provides the reasoning function with a complete view of prior interactions to guide its output.
> 3. **Construction of the task triplet.**
>    The task triplet contains $p_t$, which represents the priority level of different tasks in real-world applications. This allows AgentNet to support asynchronous and concurrent task execution. However, for a fair evaluation, we tested in the one by one method in all experiments, so this parameter was not used. We plan to explore asynchronous and concurrent execution in detail as part of future work.
>
> We sincerely appreciate the reviewer’s thoughtful comments, and in the revised version, we will provide more detailed explanations of these variables to improve the clarity of the paper.
>
> ---
>
> >**Q3.Experimental  results**
>
> **A3**
>
> We acknowledge that our method does not outperform all existing approaches on **MATH** when using **Qwen-turbo** as the backbone LLM.  However, we would like to clarify the following:
>
> 1.  **AgentNet achieves SOTA performance on other datasets and backbones.**
> With the same backbone LLM (Qwen-turbo), AgentNet reaches state-of-the-art results on the other two datasets (**BBH** and **API-Bank**).  Moreover, on the MATH dataset, AgentNet achieves or matches SOTA when using other backbones (**DeepSeek-V3** and **GPT-4o-mini**).  This demonstrates AgentNet’s strong performance in handling mathematical problems, logical reasoning, and complex tasks, as well as its excellent synergy with Qwen-turbo. We quote the original experimental results (AgentNet vs. AFlow) below:
>
> ```
> * DeepSeek-V3: 92.86% vs. 91.67% (MATH)
> * GPT-4o-mini: 85.00% vs. 85.00% (MATH)
> * Qwen-turbo: 92.00% vs. 57.00% (BBH), 32.00% vs. 22.00% (API-Bank)
> ```
>
> 2.  **Further analysis of AFlow vs. AgentNet.**
> With respect, AFLOW’s MCTS-based workflow is naturally well-suited for mathematical problems that require strictly step-by-step reasoning, which explains its relative advantage on the MATH dataset with the Qwen-turbo backbone.  However, AFLOW performs significantly worse on datasets such as BBH, which represent more diverse real-world reasoning scenarios.
>
> In addition, AFLOW need **very high computational and economic costs**, as it requires extensive LLM token consumption during workflow search and optimization.  This overhead leads to only a ~1% performance improvement.  In contrast, AgentNet achieves comparable or even superior performance using only the task execution tokens required by AFLOW, which strongly demonstrates AgentNet’s efficiency and effectiveness.
>
>
>
> ---
> >**Q4.Hyperparameter Discussion**
>
> **A4**
>
> A detailed explanation of the hyperparameters $\alpha$, $\beta$, and $\theta_{w}$ is as follows:
>
>
> - **$\alpha$ in Formula 2:**
>   $\alpha$ is a hyperparameter that balances historical information and recent interactions when updating the weights of communication edges between two agents. In AgentNet, all agents are represented as nodes in a graph, and the edge weight between two agents reflects the closeness of their relationship. After a task is executed, AgentNet updates the edge weights on the task execution chain according to whether the task was successful. $\alpha$ determines how much historical performance versus recent interaction influences this update.
>   *In our experiments, we set $\alpha$= 0.9*
>
> - **$\beta$ in Formula 10:**
>   $\beta$ is a hyperparameter that controls the update rate of an agent’s capability vector. As described in Section 3.3, when an agent completes a task and updates its capability vector, $\beta$ determines the degree of this update.
>   *In our experiments, we set $\beta$ = 0.9*
>
> - **$\theta_{w}$ in Formula 3:**
>   $\theta_{w}$is the threshold that determines whether an edge in the AgentNet network remains connected. If the weight of the edge between Agent i and Agent j falls below this threshold, the edge is pruned and the two agents are no longer considered neighbors.
>   *In our experiments, we set $\theta_{w}$ = 0.3.*
>
>
> ---
> >**Q5 & Q6. Clarity of Paper**
>
> **A5 & A6**
> We will make the necessary revisions in the updated version of the paper:
> - Add a reference to Figure 2 (System Architecture) in Section 3.2
> - Add a reference to Figure 5 (Agent Specialization) in Section 4.2
> - Add a reference to Table 2 (Heterogeneity Experiment) in Section 4.1
> - Correct the formatting error on page 8
>
> We thank the reviewer for carefully pointing out these issues regarding writing clarity.

---

> > ### Comment · Reviewer_RQrj · 2025-08-04
> >
> > Thanks for answering my concerns. I will maintain my score.

---

> > > ### Author Response · Authors · 2025-08-04
> > >
> > > We sincerely thank you for your thoughtful review, and we highly value your recognition of our work.

---

### Official Review · Reviewer_dvJt · 2025-06-30

**Clarity:** 3
**Significance:** 3
**Originality:** 2
**Rating:** 4
**Confidence:** 4

**Summary:**

This paper introduces AgentNet, a decentralized framework for coordinating multiple LLM agents through a dynamically evolving Directed Acyclic Graph (DAG). Unlike most prior multi-agent systems that rely on a central orchestrator or static roles, AgentNet allows agents to autonomously evolve, specialize, and collaborate using local decision-making, retrieval-augmented memory, and dynamic topology adaptation.

**Questions:**

1. To what extent does the performance gain stem from RAG rather than decentralization or topology learning?
2. Given that routing decisions rely on LLM decoding (prompt-based), how stable is the induced topology across repeated runs of the same task?

**Ethical Concerns:**

["NO or VERY MINOR ethics concerns only"]

**Final Justification:**

In the authors' response, most concerns have been addressed, but the author cannot provide more experimental results to support the scalability. Hence I only raise my confidence of the review.

**Limitations:**

The motivation of fully decentralized pradigm is not convincing to me, though it will be more suitable for the topology optimization.

**Quality:**

3

**Strengths And Weaknesses:**

Pros:
1. AgentNet updates the communication graph (routing edge weights) and the agent capability vectors simultaneously, based on per-task success. This allows the system to evolve a specialized, sparse DAG from an initially fully connected graph. This is a clean and elegant form of structure learning.
2. The paper includes extensive evaluations on MATH, BBH, and API-Bank with different backbones (DeepSeek, GPT-4o-mini, Qwen-turbo). The heterogeneity studies and router ablations are insightful, demonstrating how both agent diversity and router design impact overall system performance.

Cons:
1. While AgentNet claims scalability via decentralization, maintaining per-agent memory modules, capability vectors, and dynamically updated DAG connections could become computationally and memory-intensive in very large-scale systems. The actual runtime and cost implications are not analyzed.
2. For me, it is still not clear why decentralized coordination is better than a centralized controller or planner. Any experimental justification?
3. For the mult-agent baselines, it is not clear the major difference with GPTSwarm as it also share highly similar motivation in designing dynamic, specialized multi-agent systems with learnable topologies.

---

> ### Author Rebuttal · Authors · 2025-07-30
>
> >**Q1.Cost When AgentNet Scales**
>
> **A1**
>
> We sincerely thank the reviewer for raising this concern regarding scalability and for the careful evaluation of our work.
>
> 1. As shown in **Figure 7**, the performance of AgentNet continues to maintain as the multi-agent network scales up, which strongly demonstrates the usability and high performance of AgentNet under network scaling.
>
> 2. More concretely, when a round of task execution is completed and AgentNet’s parameters are updated (assuming there are k agents and each agent’s memory contains m entries, ignoring asynchronous parallelism):
>     *   **Agent memory updates**: Updating each agent’s memory requires O(m) time, resulting in a total cost of **O(km)**.
>     *   **Capability vector updates**: Each agent’s capability vector update is O(1), leading to a total cost of **O(k)**.
>     * **Topology updates**: If the task execution trajectory involves t agents (i.e., \(t-1\) edges), only the communication edges among these t agents need to be updated after the task is completed. This costs **O(t)**, and in practice, t << k.
>     * **Communication overhead**: Each agent only communicates with directly connected agents, avoiding expensive broadcast operations.
>
>    Taken together, the time complexity of AgentNet’s memory and parameter updates is only **O(km) **under large-scale applications, demonstrating its high scalability.
>
> 3. Furthermore, because of fully decentralized design, in real-world large-scale applications, each agent in AgentNet can be deployed on different devices and can asynchronously execute subtasks and update its memory and capability vectors without consuming the time or bandwidth of other devices. This greatly reduces time overhead in large-scale deployments.
>
> Due to current resource limitations, we will include a detailed comparison of LLM token consumption and other runtime costs between AgentNet and the baselines in future versions. We once again thank the reviewer for raising this important concern.
>
>
> ---
>
> >**Q2. Why decentralized better than centralized**
>
> **A2**
>
> We sincerely thank the reviewer for raising this important question regarding the advantages of decentralization. We clarify the motivation and provide experimental and theoretical justification as follows:
>
> 1. **Diversity and privacy protection.**
>    In real-world multi-agent systems, participating agents are often highly heterogeneous, belonging to different institutions or organizations, each with their own data, tools, and expertise boundaries. A decentralized framework allows each agent to maintain control over its proprietary knowledge while still contributing to collaborative problem-solving. This protects **data privacy and organizational boundaries**, which is increasingly critical in large-scale, cross-institutional deployments. In contrast, centralized controllers often require access to global information, which raises concerns over data confidentiality and ownership
>
> 2. **Broader capability coverage via collective intelligence.**
> Beyond privacy, decentralization also enables the system to fully exploit **heterogeneity at both the model and skill level**.   Each agent has its own knowledge boundaries and reasoning style;   forcing a single centralized controller to coordinate all roles inevitably biases task allocation and may underutilize agents’ specialized expertise. Decentralization, by contrast, allows agents to dynamically self‑organize and leverage complementary strengths through local decision‑making.
>
> 3. **Scalability, adaptability, and fault tolerance.**
>    Central controllers create bottlenecks (scalability), single points of failure, and rigidity in role assignment. In contrast, decentralized routing allows dynamic role reassignment and graceful handling of agent failures or overloads, improving both adaptability and robustness.
>
> 4. **Experiments on Heterogeneous Agents (Section 4.2)**
>    To specifically study the effect of diversity, Section 4.2 evaluates BBH accuracy under four settings—**Fully Homogeneous**, **Skill Heterogeneity**, **LLM Heterogeneity**, and **Both Heterogeneity** (Table 2).
>    - **3 agents:** Fully Homogeneous achieves **0.86**, while Skill/LLM/Both Heterogeneity reach **0.84 / 0.81 / 0.81**, suggesting uniform reasoning can be more effective in very small teams.
>    - **5 agents:** Heterogeneous configurations outperform the homogeneous one—Homogeneous **0.79** vs. Skill/LLM/Both Heterogeneity **0.86 / 0.85 / 0.85**—indicating that diversity enhances collaboration and complementary reasoning at larger scales.
>    These results support decentralization’s key advantage: it naturally accommodates heterogeneous agents and lets them contribute their strengths without forcing global information sharing, leading to better performance as team size grows.
>
> Decentralization addresses privacy and governance constraints while enabling dynamic adaptation and capability diversity. **Multi‑agent system research remains a rapidly evolving field, and the relative merits of centralized vs. decentralized coordination are not yet fully settled**. We will continue to explore these questions in future work. We sincerely thank the reviewer again for the thoughtful comments.
>
>
> ---
>
> >**Q3. Difference from GPTSwarm**
>
> **A3**
> AgentNet’s innovation in decentralized multi-agent systems lies in its full delegation of routing decisions to individual agents. Each agent independently determines the next step based on its own knowledge, experience, tools, and the current environment, thereby eliminating reliance on a central coordinator and achieving true decentralization. With respect, in GPTSwarm, routing decisions for each agent are made at the system level, and agents in its network do not execute sequentially;  instead, in the paper of GPTSwarm, they complete tasks individually and then engage in parallel discussions, which is fundamentally different from AgentNet's sequential execution.
>
> Moreover, GPTSwarm—as well as many existing multi-agent frameworks—still depends on a global perspective to define task paths: message-passing directions are either fixed a priori or pre-planned by a centralized planner before execution.  AgentNet, by contrast, does not rely on any global workflow or planner; each agent dynamically determines the message-passing path in real time during task execution. This design allows AgentNet to flexibly adapt to evolving environments without requiring a static or planner-driven topology.
>
>
>
> ---
>
>
> > **Question 1. To what extent does the performance gain stem from RAG rather than decentralization or topology learning?**
>
> As shown in **Table 1**, the Synapse baseline represents a single-agent system that adopts both a ReAct-based architecture and Retrieval-Augmented Generation (RAG), but without decentralization or dynamic topology learning. Compared with other single-agent baselines without RAG, Synapse performs noticeably better, highlighting the substantial contribution of RAG to task execution performance.
>
> However, AgentNet also adopts a ReAct-style agent architecture and RAG, in addition to decentralization and dynamic topology learning. The further performance gains observed in AgentNet over Synapse indicate that RAG alone cannot fully account for the improvements. Instead, these results suggest that decentralization and learned topological routing provide complementary benefits by enabling more scalable coordination and more adaptive task delegation.
>
> ---
>
> > **Question 2. Given that routing decisions rely on LLM decoding (prompt-based), how stable is the induced topology across repeated runs of the same task?**
> Due to the high cost of repeated API calls to large language models, we were unable to conduct large-scale repeated trials to systematically assess topology stability. However, based on our practical observations, we can share the following findings:
>
> AgentNet's routing decisions are primarily influenced by two factors: (1) the agent's current estimated capabilities, and (2) routing history, including past successes or failures with specific neighbors. We found that when the same task is repeated, the system tends to reuse successful routes or avoid previously unsuccessful ones, reflecting a form of empirical consistency in routing behavior.
>
> In ambiguous situations (e.g., when multiple neighboring agents have identical capabilities, success rates, and edge weights), the LLM sometimes chooses the agent with the smallest ID. This is typically explained by the LLM as a simple preference rather than stochasticity. We observed this behavior did not negatively impact task performance.
>
> Importantly, all experiments were run with temperature set to 0, ensuring deterministic LLM decoding. This minimizes variability due to sampling randomness and further stabilizes the induced topology. Together, these observations suggest that routing decisions in AgentNet, though prompt-based, exhibit a high degree of stability under repeated executions.
>
> ---
>
> In summary, we believe the above findings sufficiently address the reviewer’s questions.

---

> > ### Author Response · Authors · 2025-08-04
> >
> > Thank you for your thoughtful review. We have carefully addressed the concerns you mentioned. We are open to any further discussion or clarification！

---

> > ### Comment · Reviewer_dvJt · 2025-08-06
> >
> > Thanks for the authors' reply. I will raise my confidence with the orginal positive score.

---

> > > ### Author Response · Authors · 2025-08-08
> > >
> > > Thank you for acknowledging AgentNet! We truly appreciate this opportunity to clarify our contributions during the rebuttal process.

---

### Official Review · Reviewer_9hci · 2025-07-03

**Clarity:** 3
**Significance:** 3
**Originality:** 3
**Rating:** 5
**Confidence:** 4

**Summary:**

This paper proposes AgentNet, a decentralized framework for LLM-based multi-agent systems (MAS). It addresses limitations of centralized architectures (e.g., scalability bottlenecks, single points of failure) by enabling agents to coordinate via a dynamic Directed Acyclic Graph (DAG) with retrieval-augmented generation (RAG). Key innovations include: 1) fully decentralized task routing without a central controller; 2) adaptive graph topology that evolves based on task demands; 3) RAG-based memory for continuous skill refinement. Experiments show AgentNet outperforms single-agent and centralized MAS baselines in task accuracy, especially in dynamic environments. It also demonstrates robust privacy preservation and cross-organizational collaboration via localized data processing.

**Questions:**

See `Weaknesses`.

Overall, I acknowledge the substantial engineering effort behind this work, so I would lean toward accepting the paper.

**Ethical Concerns:**

["NO or VERY MINOR ethics concerns only"]

**Final Justification:**

The author's response resolved my concerns about the method's innovation, experimental results, and training costs. So I raised the score from 4 to 5 accordingly.

**Limitations:**

yes

**Quality:**

3

**Strengths And Weaknesses:**

**Strengths**

+ Novel Decentralized Design: Eliminates central orchestrators, enhancing fault tolerance and scalability. The dynamic DAG allows agents to self-organize, addressing rigidity in traditional MAS.

+ Adaptive Learning Mechanisms: RAG-based memory and evolutionary task routing enable agents to specialize over time, improving performance in complex tasks.

+ Dynamic agent graph topology

+ The paper is characterized by a clear structure and standardized academic writing.

**Weaknesses**

The paper is generally well-executed, with a clear algorithmic description. However, I have the following concerns:

* **Similarity to existing methods**: Some modules resemble prior work. For example, the *Decentralized Network Topology* is highly similar to GPTSwarm and AgentPrune. The idea of *Adaptive Learning and Specialization*, based on RAG, has been explored in many previous studies and appears quite similar to the approach in ExpeL. While I acknowledge the authors' contributions to decentralization, it seems that decentralized ideas have also been adopted in dialogue-driven systems like AutoGen.

* **Concerns about empirical results**: Compared to the AFlow algorithm proposed in 2024, the performance improvements of AgentNet do not appear to be particularly significant.

* **Questions about training cost**: Perhaps I missed it, but I’m curious about the training overhead of AgentNet. Compared to AFlow, this might be one of AgentNet's advantages.

---

> ### Author Rebuttal · Authors · 2025-07-30
>
> >**Q1.Comparison With Existing Methods**
>
> **A1**
> We sincerely thank the reviewer for raising the concern regarding the similarity of some modules with prior work. While we share a common motivation with earlier efforts in building robust multi-agent systems, AgentNet **introduces unique innovations** that set it apart from existing approaches.
>
> 1. **Our innovation in decentralized multi-agent systems**:
>
>     Our fully decentralized design stems from practical considerations. As more companies develop their own agents, we foresee agents interacting across the internet in a decentralized manner, making fully decentralized MASs increasingly necessary. AgentNet delegates routing decisions to each independent agent, allowing them to adaptively make routing choices based on dynamic environments, thereby removing the reliance on a central coordinator.
>
>     In contrast, GPTSwarm determines routing decisions at the system level, and its agents do not execute sequentially but rather perform tasks individually and then engage in parallel discussions. This is fundamentally different from AgentNet’s sequential execution. AgentPrune, on the other hand, adopts a one-shot pruning method with a relatively complex optimization process, and it lacks adaptability to dynamic real-world environments. Furthermore, unlike frameworks like AutoGen that focus on fixed workflows and predefined message paths, AgentNet enables agents to autonomously decide routing decisions, based on their own and their neighbors' state and expertise. This results in a more flexible, decentralized approach to task allocation.
> 2. **Why RAG is applied in AgentNet**:
>     AgentNet highlights the importance of private knowledge and experience updates for each agent. In real-world deployments, each agent in a multi-agent system may have its own role, along with private databases, tools, and other unique resources. The quality and domain of these databases directly influence the agent’s capability through training parameters or demonstrations. In AgentNet, we model this reality using a RAG-based mechanism, enabling each agent to maintain distinct knowledge, perform unique updates, and retrieve information effectively.
>
> We appreciate the reviewer’s thoughtful concern, and we will include a more detailed clarification of these points in the next revision.
>
>
>
> ---
>
> >**Q2. Experimental results**
>
> **A2**
>
> We acknowledge that our method does not outperform all existing approaches on the MATH dataset when using Qwen-turbo as the backbone LLM. However, we would like to clarify the following points:
>
> 1. **State-of-the-art performance across multiple datasets and backbones.**
> With the same backbone LLM (Qwen-turbo), AgentNet achieves SOTA performance on the other two datasets (BBH and API-Bank). Moreover, on the MATH dataset, AgentNet also reaches or surpasses SOTA performance when using other backbone LLMs (DeepSeek-V3 and GPT-4o-mini). This demonstrates AgentNet’s outstanding ability compared to other baselines in handling mathematical problems, logical reasoning, and other complex tasks, as well as its strong synergy with Qwen-turbo. We quote the original experimental results (AgentNet vs. AFlow) below:
>
> ```
>     * DeepSeek-V3: 92.86% vs. 91.67% (MATH)
>     * GPT-4o-mini: 85.00% vs. 85.00% (MATH)
>     * Qwen-turbo: 92.00% vs. 57.00% (BBH), 32.00% vs. 22.00% (API-Bank)
> ```
>
>
> 2. **Further analysis of AFlow vs. AgentNet.**
>     With respect, AFlow’s MCTS-based workflow is naturally well-suited for mathematical problems that require step-by-step reasoning, which explains its relatively strong performance on the MATH dataset. However, on BBH, which represents more diverse real-world logical reasoning tasks, AFlow’s performance is comparatively weaker.
>
>     Furthermore, AFlow need a substantial computational and economic cost, as it requires a large number of LLM tokens during workflow search and optimization. This heavy cost results in only a ~1% performance gain on MATH. In contrast, AgentNet achieves comparable or even superior performance using only the task execution tokens required by AFlow, demonstrating its efficiency and effectiveness.
>
>
>
>
> ---
>
> >**Q3. Concerns about empirical results**
>
> **A3**
> Thank you for raising this excellent question!
>
> At a high level, both AgentNet and AFlow require a warm-up phase on a training dataset disjoint from the test set, followed by responding to each query in the test set.
>
> * AgentNet consists of 3 agents. For each query, each agent performs the equivalent of 2 LLM calls and 4 memory retrievals (1 LLM call and 2 memory retrievals each for the router and executor). AgentNet performs 1 warm-up round on the training dataset.
> * AFlow, during warm-up, requires on average 6 LLM calls per query per round. It performs 20 warm-up rounds on the training dataset, with its best performance usually reached at round 5.
>
> Compared with AFlow, the major cost of AgentNet lies in memory construction during initialization. However, its runtime cost is significantly lower because:
> ```
> 1. It does not need to maintain a global state
> 2. Its decisions are localized, resulting in lower communication overhead
> 3. The dynamic pruning mechanism effectively controls memory growth
> ```
>
> In the warm-up stage, AgentNet only needs to roll out one round on the warm-up dataset, instead of exploring the large search space that AFlow requires. Moreover, our approach is designed to be deployment-friendly: the running network state itself naturally aligns with the warm-up state.

---

> > ### Comment · Reviewer_9hci · 2025-08-03
> >
> > Thanks for the reply. The authors' reply solved all my doubts so I will increase my score from 4 to 5.

---

> > > ### Author Response · Authors · 2025-08-04
> > >
> > > Thank you for your positive feedback and for recognizing the value of our work. We truly appreciate your support！

---

### Note · Authors · 2025-08-13

# Author Final Remarks

We express our sincere gratitude to the reviewers for their time and feedback.   We've carefully addressed all raised concerns and made revisions to clarify key aspects of our work.

## Summary of Revisions:

### 1. Clarification of Decentralization and Topology Learning:
We’ve elaborated on how the decentralized approach, dynamic agent graph topology, and retrieval-augmented memory mechanism work together to improve performance.   We clarified that decentralization and dynamic agent self-organization enable specialization, enhancing scalability and adaptability.

### 2. Addressing Hallucination and Inconsistencies:
In response to concerns about hallucinations and inconsistencies, we clarified that AgentNet employs two mechanisms: a verification process where agents cross-check each other’s outputs to minimize errors, and a specialization process where agents, as they become more specialized over time, reduce the likelihood of errors in their specific tasks.

### 3. Improving Writing Clarity:
Terminology has been refined to clarify the distinctions between key concepts like “capabilities” vs. “abilities,” and contrasts such as “static” vs. “adaptive” and “hierarchical” vs. “decentralized.”

### 4. Experimental Methodology and Hyperparameters:
We provided clearer explanations of how agent capabilities evolve and the relationship between context and capability vectors.   We also clarified the hyperparameters used in experiments.

### 5. Handling Cyclic Dependencies:
To address concerns about cyclic dependencies, we clarified that the directed acyclic graph (DAG) structure ensures one-way task execution, preventing infinite loops.

### 6. Scalability and Computational Complexity:
We discussed how AgentNet's decentralized design scales efficiently with minimal computational overhead, highlighting its comparative advantage in computational cost.

## Final Thoughts:
We believe these revisions address all concerns and clearly demonstrate the contributions of AgentNet.   Our work presents significant advancements in decentralized multi-agent systems, offering a flexible, scalable solution for future research.

We are grateful for the opportunity to revise the paper and welcome further discussions to enhance its impact.   Thank you again for your consideration.

---

### Decision · Program_Chairs · 2025-09-17

**Decision:**

Accept (poster)

**Comment:**

### (a) Summary of Claims
This paper introduces AgentNet, a novel decentralized framework for LLM-based multi-agent systems (MAS). It aims to overcome the limitations of centralized architectures, such as scalability bottlenecks and single points of failure, by enabling agents to coordinate autonomously through a dynamic Directed Acyclic Graph (DAG). The framework integrates a Retrieval-Augmented Generation (RAG) based memory system to allow agents to continuously learn and specialize over time. The authors claim that this decentralized, evolutionary approach improves task accuracy, adaptability, and scalability, particularly in dynamic environments, outperforming single-agent and centralized MAS baselines.

### (b) Strengths
* Novel and well-motivated architecture design.
* Adaptive learning and structure.
* Comprehensive evaluation.

### (c) Weaknesses
* Scalability: The current manuscript lacks large-scale empirical experiments to fully validate its claims.
* Limited evaluation: Only a limited number of tasks were evaluated.

### (d) Reasons for acceptance
This paper presents a robust and practical implementation of a decentralized autonomous multi-agent system, backed by a comprehensive evaluation. While the core idea is not novel, the AC recommends acceptance due to the importance of highlighting decentralized autonomous multi-agent systems within the multi-agent community.